# Imaging nodal knots in momentum space through topolectrical circuits

Ching Hua Lee 1✉, Amanda Sutrisno 2, Tobias Hofmann[3], Tobias Helbig[3], Yuhan Liu[4,5], Yee Sin Ang 2✉, Lay Kee Ang 2, Xiao Zhang 5✉, Martin Greiter 3 & Ronny Thomale 3✉

Knots are intricate structures that cannot be unambiguously distinguished with any single topological invariant. Momentum space knots, in particular, have been elusive due to their requisite finely tuned long-ranged hoppings. Even if constructed, probing their intricate linkages and topological "drumhead" surface states will be challenging due to the high precision needed. In this work, we overcome these practical and technical challenges with RLC circuits, transcending existing theoretical constructions which necessarily break reciprocity, by pairing nodal knots with their mirror image partners in a fully reciprocal setting. Our nodal knot circuits can be characterized with impedance measurements that resolve their drumhead states and image their 3D nodal structure. Doing so allows for reconstruction of the Seifert surface and hence knot topological invariants like the Alexander polynomial. We illustrate our approach with large-scale simulations of various nodal knots and an experiment which maps out the topological drumhead region of a Hopf-link.

[1] Department of Physics, National University of Singapore, Singapore 117542, Singapore. [2] Science, Mathematics and Technology, Singapore University of Technology and Design, Singapore 487372, Singapore. [3] Institute for Theoretical Physics and Astrophysics, University of Würzburg, Am Hubland, Würzburg D-97074, Germany. [4] Department of Physics, The University of Chicago, Chicago, IL 60637, USA. [5] School of Physics, Sun Yat-sen University, Guangzhou 510275, China. ✉email: phylch@nus.edu.sg; yeesin_ang@sutd.edu.sg; zhangxiao@mail.sysu.edu.cn; rthomale@physik.uni-wuerzburg.de

In the pursuit of ever more exotic topological states, contemporary research has witnessed a shift from established topological insulator platforms with $\mathbb{Z}$ or $\mathbb{Z}_2$ topology to photonic, mechanical, and acoustic metamaterials[1–3] that mimic topological nodal semimetals[4–10]. The conceptual transfer from conventional electronic materials to such artificial structures allows for unprecedented control over individual couplings, and further permits access to any spectral regime of the band structure without limitations, as, e.g., implied by the chemical potential for electronic matter. The recent introduction of electric circuits for topological engineering[11–17] brought about even greater accessibility and fine tuning, as well as much reduced cost. Most importantly, however, circuit connections transcend locality and dimensionality constraints, putting the implementation of couplings between distant sites of a high-dimensional system and nearest-neighbor connections on equally accessible footing. Furthermore, density of states divergences[18] and even admittance bandstructure[15,19] can be obtained with just impedance and voltage/current measurements, respectively.

Among topological structures, knots rank as among the most exotic, being intimately connected to Chern-Simons theory which underlies the braiding of quasiparticles[20,21]. In real space, knots are ubiquitous, being present in protein and polymer structures, optical vortices[22] and, of course, everyday-life ropes. In momentum space, knotted configurations of band structure crossings (nodes) demonstrate their topological intricacies even more spectacularly, with their special "drumhead" surface modes generalizing the Fermi arcs of ordinary nodal semimetals.

To realize and image momentum space nodal knots in RLC circuits, two challenges have to be overcome. First, RLC circuits are reciprocal due to their components being symmetric from both ends, but mathematical models of nodal knots proposed thus far[23–27] imply broken reciprocity. This apparent limitation has prevented nodal knot circuits from being developed so far, despite successes in non-knotted nodal loop circuits and metamaterials[28–31]. Second, the momentum knots are subextensive 1D features of the 3D Brillouin zone (BZ), and great finesse is required in imaging them.

In this work, we show how these challenges can be overcome via (i) a special scheme for designing nodal knots circuits with mirror-image partners, (ii) a new robust impedance measurement approach for imaging nodal knots and their accompanying drumhead surface states, and (iii) an instructive experimental demonstration of how the topological drumhead region of a nodal knot can imaged.

## Results

**Designer nodal knots from braids.** The most natural route to realizing momentum space knots is via a 3D lattice with band intersections (nodes) along particular knotted trajectories. A generic reciprocal lattice with band intersections minimally contains two sites per unit cell, and can be written as a reciprocal (momentum) space graph Laplacian

$$J(\mathbf{k}) = l_0 \, \mathbb{I} + \Re f(\mathbf{k})\tau_x + \Im f(\mathbf{k})\tau_z, \qquad (1)$$

where $l_0$ is a uniform offset, $f(\mathbf{k})$ is an even function of $\mathbf{k}$, and $\tau_x$, $\tau_z$ are the Pauli matrices. Nodes occur whenever its two eigenvalues (bands) $l_0 \pm \sqrt{[\Re f(\mathbf{k})]^2 + [\Im f(\mathbf{k})]^2} =: l_0 \pm |f(\mathbf{k})|$ coincide, i.e., yielding a vanishing gap $2|f(\mathbf{k})| = 0$. This is a complex constraint equivalent to the intersection of two level sets given by $\Re f(\mathbf{k}) = 0$ and $\Im f(\mathbf{k}) = 0$, which hence traces out a 1D nodal line in the 3D BZ. Note that we have excluded $\tau_y$ terms, which will break the nodal line into isolated Weyl points. Generically, the locus of $f(\mathbf{k}) = 0$ can correspond to broken arcs or arbitrarily intertwined closed loops. The topologically most

interesting cases occur when a loop links nontrivially with itself, forming a nodal knot, or when multiple loops inseparably entangle to form a nodal link. In the following, we shall first show how $f(\mathbf{k})$ can be constructed based on a desired knot or link structure, without restricting ourselves to any particular physical implementation. Subsequently, we show why its corresponding Laplacian $J(\mathbf{k})$ can be most suitably implemented by an RLC circuit.

To design $f(\mathbf{k})$, the first step is to unambiguously specify a desired knot or link. Intuitively, we can visualize a knot/link as a braid closure[32], i.e., as a collection of intertwining strands with their permuted ends joined together. (Fig. 1: The number of linked components is equal to the number of cycles in the decomposition of the permutation.) The precise sequence of the strand crossings identifies the knot/link, and is annotated as a braid word $\sigma_1^{\pm} \sigma_2^{\pm} \ldots$, with $\sigma_i$ indicating that the $i^{\text{th}}$ string crosses above the $(i+1)^{\text{th}}$ string from the left, and $\sigma_i^{-1}$ if the crossing is from below. Two non-adjacent crossings commute: $\sigma_i\sigma_j = \sigma_j\sigma_i$ for $|i - j| \geq 2$; less obvious is the braid relation $\sigma_i\sigma_j\sigma_i = \sigma_j\sigma_i\sigma_j$ which plays a fundamental role in the Yang–Baxter equation[33]. Note that due to the braid relation, as well as Markovian moves that swap the closing strands[34], more than one braid word can correspond to a desired knot. Nevertheless, the specification of the braid uniquely identifies the knot. For instance, $\sigma_1^2$ gives the Hopf-link, while $\sigma_1^3$ gives the Trefoil knot (Fig. 1).

The next step is to find an explicit form of $f(\mathbf{k})$ that gives the knot/link corresponding to a desired braid. Mathematically, the knot/link exists as the kernel of the mapping $f : \mathbb{T}^3 \to \mathbb{C}$, which maps $\mathbf{k}$ in the 3D BZ $\mathbb{T}^3$ onto a complex number $f(\mathbf{k})$. To make sure that $f$ incorporates the information from the braid, we decompose it into a composition of mappings

$$\mathbb{T}^3 \xrightarrow{F} \mathbb{C}^2 \xrightarrow{\bar{f}} \mathbb{C}, \qquad (2)$$

i.e., $f(\mathbf{k}) = \bar{f}(F(\mathbf{k}))$ where $F(\mathbf{k}) = (z, w)$ maps $\mathbf{k}$ onto two complex numbers $z(\mathbf{k})$ and $w(\mathbf{k})$ in an auxiliary braiding space, which then yields $f$ via the braiding map $\bar{f}(z(\mathbf{k}), w(\mathbf{k})) = f(\mathbf{k})$. To concretely understand this decomposition, we first note that a braid closure lives in the space $\mathbb{C} \times S^1$, since the position of $N$ strands can be given by complex coordinates $z_1(s), z_2(s), \ldots, z_N(s)$, where $s \in [0, 2\pi]$ is the periodic vertical "time" coordinate (Fig. 1a). Each braid operation corresponds to two half-revolutions (windings) between two particles i.e. $\sigma_i^{\pm}$ corresponds to $z_{i+1} - z_i \to e^{\pm i\pi}(z_{i+1} - z_i)$ with increasing $s$. We thus define $\bar{f}(z, w)$ by analytical continuation to complex $s = -i\log w$ as

$$\bar{f}(z, e^{is}) = \prod_j^N \left( z - z_j(s) \right), \qquad (3)$$

such that points satisfying the nodal constraint $\bar{f}(z, w) = 0$ lie exactly along the trajectories $z_j(s)$. To use Eq. (3), one expresses each $z_j(s)$ as a time Fourier series containing $w = e^{is}$, i.e., a polynomial in $w$, such that $\bar{f}(z, w)$ becomes a Laurent polynomial of $z$ and $w$. For instance, a Hopf braid can be parametrized by $z_1(s) = -z_2(s) = e^{is} = w$, which yields $\bar{f}(z, w) = (z - w)(z + w) = z^2 - w^2$. This can be directly generalized to a braid of a $(p, q)$ torus knot, which consists of $p$ strands each of which twists for $q$ revolutions before closure: $z_j(s) = e^{\frac{i}{p}(2\pi j + qs)}$, yielding $\bar{f}(z, w) = z^p - w^q$. Next, we need a criterion for suitable functions $F(\mathbf{k}) = (z(\mathbf{k}), w(\mathbf{k}))$, that express $z$ and $w$ in terms of $\mathbf{k}$. Ideally, $F(\mathbf{k})$ should be able to "curl up" the braiding space $\mathbb{C} \times S^1$ into a solid torus in the 3D BZ, such that knots given by braid closures are faithfully mapped into nodal knots in the 3D BZ[35] (Fig. 1). How this "curling" is accomplished is

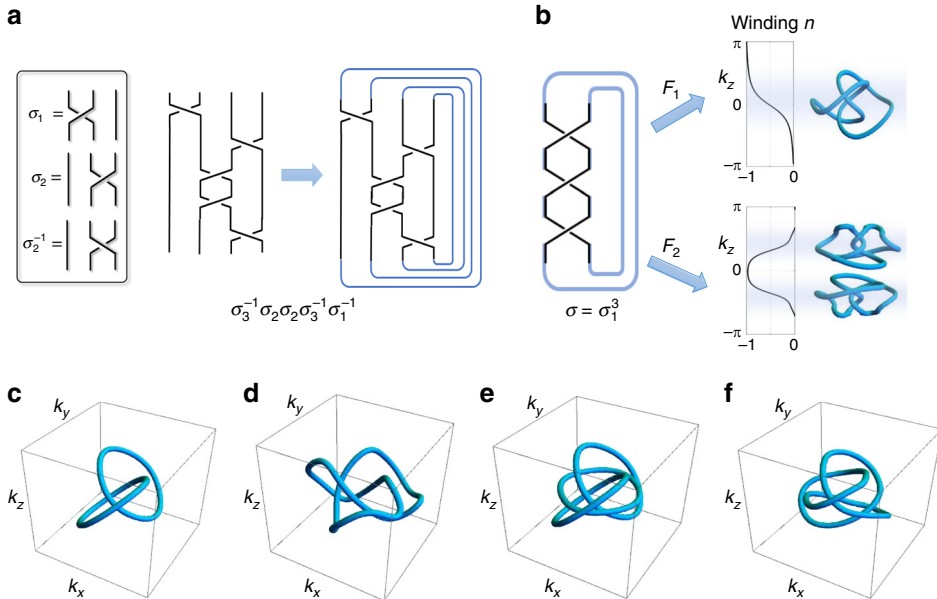

**Fig. 1 Nodal knots from braids. a** Braid operations $\sigma_i$ and $\sigma_i^{-1}$ represent the over/under-crossing of strand $i$ with strand $i+1$ as we travel upwards. A braid consists of a series of braid operations, and can be closed to form a knot or link (in this case it is a link between three loops). **b** A braid closure can be embedded onto the 3D BZ torus in different ways through different choices of $F(\mathbf{k})$. Depending on its topological charge density distribution of Eq. (4), it can produce different numbers of copies of the knots in the BZ, i.e. one a single copy ($F_1$) or two mirror imaged copies ($F_2$). **c–f** Various examples of simple Nodal knots/links defined by Eq. (3), some of which we shall explicitly construct in circuits band structures later. **c** Hopf-link with $\sigma = \sigma_1^2$ and $f(z, w) = (z - w)(z + w)$. d) Trefoil knot with $\sigma = \sigma_1^3$ and $f(z, w) = (z - w^{3/2})(z + w^{3/2})$. e) 3-link with $\sigma = (\sigma_1 \sigma_2 \sigma_1)^2$ and $f(z, w) = z(z^2 - w^2)$. **f** Figure-8 knot with $\sigma = (\sigma_2^{-1} \sigma_1)^2$ and $f(z, w) = 64z^3 - 12z(3 + 2(w^2 - \bar{w}^2)) - 14(w^2 + \bar{w}^2) - (w^4 - \bar{w}^4)$[35].

quantified by the winding number

$$n = -\frac{1}{2\pi^2} \int_{BZ} d^3\mathbf{k}\, \epsilon_{\mu\nu\rho\gamma} N_\mu \partial_{k_x} N_\nu \partial_{k_y} N_\rho \partial_{k_z} N_\gamma, \quad (4)$$

where $\mu, \nu, \rho, \gamma \in \{1, 2, 3, 4\}$ and $z(\mathbf{k}) = N_1(\mathbf{k}) + iN_2(\mathbf{k})$, $w(\mathbf{k}) = N_3(\mathbf{k}) + iN_4(\mathbf{k})$. It measures how many times the braid winds around the BZ. Generically, one will choose an $F(\mathbf{k})$ with winding $n = \pm 1$ to guarantee a one-to-one mapping from a specific braid closure to a nodal knot in the BZ. An important caveat, however, is that $n = \pm 1$ is not possible for a passive RLC circuit implementation due to its reciprocal nature. In the discussion surrounding Eq. (7) later, we shall explain how this seeming obstacle can be avoided systematically.

Our approach outlined so far generalizes existing approaches in the literature: In the approach of Ezawa[23], $F(\mathbf{k})$ was chosen to be certain generalized Hopf fibrations, but there was no freedom of choosing $f(z, w)$ for more general knot constructions; $f(z, w)$ was further explored in ref. [36] in real space, but not in a toroidal momentum BZ where a nodal bandstructure can be found.

**Characterizing nodal knot topology**. A key feature of nodal knots is their interesting topological structure. Knotted lines of singularities in momentum space can be viewed as generalizations of Weyl points. In place of isolated sources of topological (Berry) flux, there are intertwined loops of "branch cuts". While signatures of non trivial knot topology can manifest as optical non-linearity enhancements in electronic nodal materials[37,38], we shall see that circuit implementations allow the nodal knots themselves to be directly reconstructed.

To mathematically characterize different knots, we first introduce the knot group. The knot group of a given knot $K$ is the fundamental group $\pi_1(\mathbb{T}^3 \setminus K)$ of its complement in its ambient space, which in our context is the 3-torus BZ $\mathbb{T}^3$. Physically, the complement $\mathbb{T}^3 \setminus K$ is the part of the BZ containing non-degenerate eigenmodes, and the knot group

indexes the space of non-trivial closed paths within this phase space. In the simple case of a nodal ring (unknot), $\pi_1(\mathbb{T}^3 \setminus K)$ consists of equivalence classes of trajectories characterized by their winding number around the ring, and is thus given by integer-valued Berry phase windings $\mathbb{Z}$. In more complicated knots, there can be several inequivalent sets of windings, corresponding to different unique homotopy generators of $\mathbb{T}^3 \setminus K$. For instance, the knot group of a $(p, q)$ torus knot is given by $\langle x, y | x^p = y^q \rangle$, since a path that winds $p$ times around the "equator" can be deformed into one that winds $q$ times around the "pole". In the special case of the trefoil knot with $(p, q) = (2, 3)$, the knot group $\langle x, y | x^2 = y^3 \rangle$ is also isomorphic to the braid group with three strands: $\sigma_1 \sigma_2 \sigma_1 = \sigma_2 \sigma_1 \sigma_2$, as evident from identifying $x = \sigma_1 \sigma_2 \sigma_1$ and $y = \sigma_1 \sigma_2$. Yet, in general, the presentation for the knot group can take diverse reparametrized forms (i.e. $\langle x, y | xyx^{-1}yx = yxy^{-1}xy \rangle$ for the figure-8 knot), and is hence by itself insufficient for topological classification.

In order to faithfully distinguish topologically inequivalent knots, various knot invariants have been developed. Simple invariants such as the linking number or knot signature can be easily computed by examining the crossings, but only have limited discriminatory power. A more sophisticated approach involves the Chern Simons path integral[20], which encapsulates topological information on the nodal singularities through certain knot polynomials, i.e., Jones polynomials, depending on the chosen gauge group. In our physical setup with classical circuits, another well-established invariant known as the Alexander polynomial will be most experimentally accessible. Starting from the topological surface "Drumhead" modes, one can reconstruct the Seifert surface, which is an orientable surface in the 3D BZ whose boundary is the nodal knot/link, and compute the Alexander polynomial from its homology properties.

**Surface states of knots**. Since nodal knots/links consist of closed loops, they form the boundary of topological surface drumhead

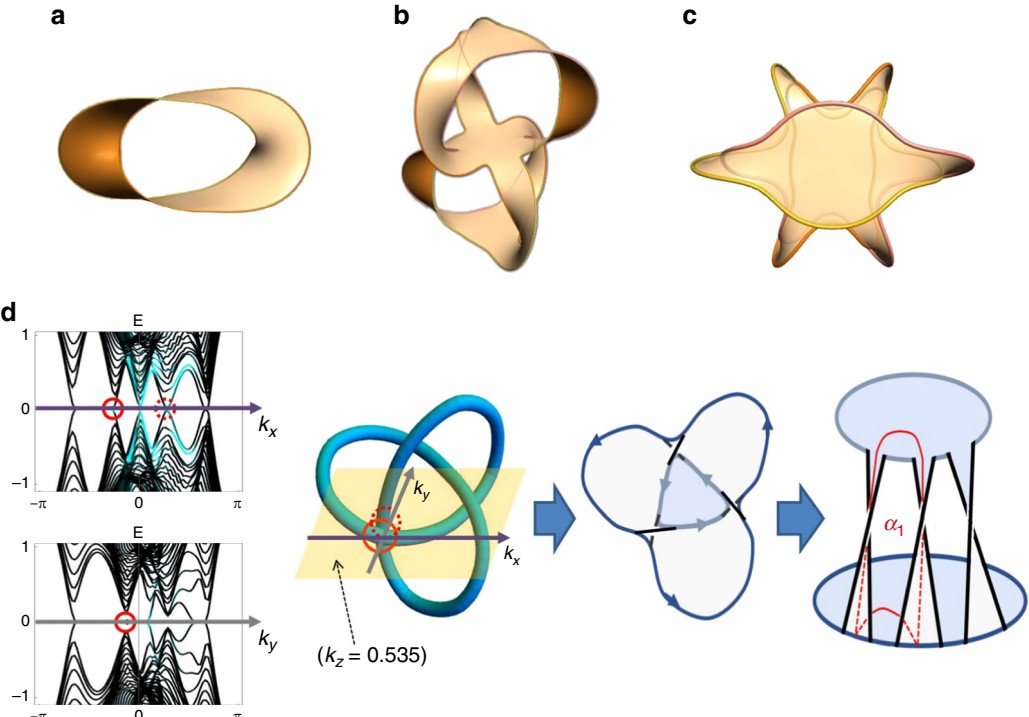

**Fig. 2 Seifert surfaces from topological surface states.** Projected surface states on the (001) surface of the **a** Hopf-link with $\sigma = \sigma_1^2$, **b** Borromean rings with $\sigma = (\sigma_2^{-1}\sigma_1)^3$ and c) 3-link with $\sigma = (\sigma_1\sigma_2\sigma_1)^2$. We can observe multiple folded layers of the surface on top of another. Note that a different parametrization was used to plot these surfaces, as compared to Fig. 1. Interestingly, **b**, **c**, both contain three loops, but **b** is totally unlinked upon removal of any single loop, while, **c** still reduces to a Hopf-link upon removal of any loop. **d** How a Seifert surface can be obtained from the Drumhead states. By comparing the same nodal crossings across Drumhead states from different surfaces (Left), one can deduce the over/under-crossings in a knot diagram. The interior of this knot can then be systematically promoted into "surface layers" bounded by appropriately defined crossings (Center), which can further be arranged into a layer arrangement where its homology loops (i.e., $\alpha_1$) are evident.

modes in the projected 2D surface BZ. Intuitively, drumhead modes can be construed as Fermi arcs traced out by Weyl points moving along the nodal lines. If a nodal structure were to be deformed across a topological transition, i.e., till the loops of a Hopf link intersect, the shape of the drumhead regions along suitable projections must also transition discontinuously i.e. from two overlapping regions to two disjoint regions. For each possible surface termination, the drumhead regions form the surface projections (shadow) of a tight, i.e., minimal area Seifert surface (Fig. 2). In this sense, the drumhead modes on differently oriented boundary surfaces are just different "holographic" projections of the same tight Seifert surface living in the 3D BZ. Note that a Seifert surface is itself not a topological invariant, since it is not unique: for instance, $\text{Re}[f(\mathbf{k})] > 0$, $\text{Re}[f(\mathbf{k})] < 0$, $Im[f(\mathbf{k})] > 0$ and $Im[f(\mathbf{k})] < 0$ are all valid Seifert surfaces, albeit not all tight.

To construct a topological invariant such as the Alexander polynomial, we hence need information on how the Seifert surface links with itself: we consider the linking of its 1st-homology loops $\alpha_1, \alpha_2, \ldots, \alpha_l$ with $\alpha_1', \alpha_2', \ldots, \alpha_l'$ of a lifted Seifert surface defined from a infinitesimally shifted Laplacian $L'(\mathbf{k}) = L(\mathbf{k}) - \epsilon\tau_j$, with $j = x$ or $z$. This shift creates a parallel Seifert surface infinitesimally displaced in a way consistent with the knot orientation given by the vector $\nabla_{\mathbf{k}}\text{Re}\, f(\mathbf{k}) \times \nabla_{\mathbf{k}}Im\, f(\mathbf{k})$. The $l \times l$ Seifert matrix $S_{ij}$, which captures the twisting structure of the Seifert surface, is then given by the linking number of $\alpha_i$ and $\alpha_j'$, with $l$ being the number of homology generators[34,39]. From that, one can obtain the Alexander polynomial invariant as

$$A(t) = t^{-l/2}\text{Det}[S - tS^T]. \tag{5}$$

For instance, as further elaborated on in the methods section, $A$

$(t) = t + t^{-1} - 1$ for the trefoil knot. General heuristics for constructing and visualizing the Seifert surface for a given nodal bandstructure are outlined in Fig. 2d.

**Constructing and measuring knots in circuits.** Having detailed their mathematical construction and characterization, we now describe how nodal knots can be concretely implemented and detected in electrical RLC circuits via both simulations and experiments. An RLC circuit with $N$ nodes can be represented by an undirected network with graph nodes (junctions) $\alpha = 1, \ldots, N$ connected by resistors, inductors and capacitors. Its behavior is fully characterized by Kirchhoff's law at each junction, which takes the matrix form

$$I_\alpha = J_{\alpha\beta}V_\beta, \tag{6}$$

where $I_\alpha$ is the external current entering junction $\alpha$ and $V_\beta$ is the potential at junction $\beta$. Physically, each entry $J_{\alpha\beta}$ of the Laplacian $J$ physically represents admittance (AC conductance): in the submatrix spanned by junctions $(\alpha, \beta)$, an element with impedance $r_{ab}$ contributes $r_{ab}^{-1}(1\; -1\; -1\; 11)$ to the Laplacian, where $r_{ab} = R$, $i\omega L$ and $(i\omega C)^{-1}$ for the RLC components, respectively. The strictly reciprocal (symmetric) nature of these components constrains the possible forms of the Laplacian. In particular, for a circuit array with two sites per unit cell, $\text{Re}\, f(\mathbf{k})$ and $Im\, f(\mathbf{k})$ in the Laplacian of Eq. (1) must be even[40] in powers of $\mathbf{k}$. This constraint severely restricts the prospects of faithfully "curling" a braid into a 3D BZ, such that each desired braid crossing is mapped one-to-one onto the resultant nodal structure. This is because nodal knots necessarily contain unpaired 2D Chern phase slices, which require reciprocity breaking. Mathematically, it corresponds to the impossibility of achieving an $F(\mathbf{k})$ winding

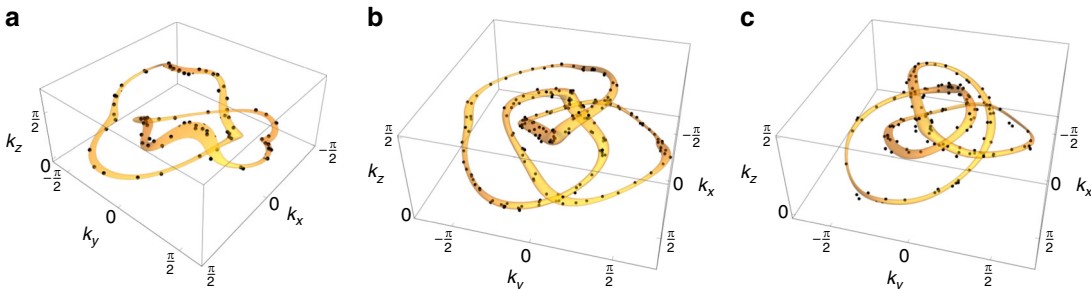

**Fig. 3 Simulated nodal structure measurements under PBCs.** Points in reciprocal space corresponding to admittance eigenvalues smaller than a threshold $j_s$ are colored black, which collectively delineate their theoretically computed respective nodal links or knots (orange). **a** Two entangled unknots, defined as the Hopf-link. The black dots combine the simulation results with circuit dimensions of $(22 \times 22 \times 16)$, $(23 \times 23 \times 20)$, $(16 \times 22 \times 19)$, $(22 \times 22 \times 14)$, $(25 \times 20 \times 23)$ and $(25 \times 24 \times 23)$. The admittance threshold is chosen to be $j_s = 0.00335 \, \Omega^{-1}$. **b** Depicts a trefoil knot showing the combined simulations of circuit system sizes of $(20 \times 20 \times 20)$, $(21 \times 21 \times 21)$, $(24 \times 15 \times 15)$, $(21 \times 20 \times 25)$, $(18 \times 19 \times 17)$, $(17 \times 18 \times 21)$, $(23 \times 21 \times 19)$, $(19 \times 25 \times 23)$ and $(20 \times 20 \times 22)$. The admittance bound is threshold $j_s = 0.0032 \, \Omega^{-1}$. **c** Illustrates a figure-8 knot with $(23 \times 23 \times 23)$, $(20 \times 20 \times 25)$, $(20 \times 20 \times 21)$, $(19 \times 16 \times 18)$, $(17 \times 14 \times 16)$, $(19 \times 25 \times 25)$ and $(25 \times 21 \times 22)$ unit cells in the respective directions. The admittance threshold is chosen to be $j_s = 0.0037 \, \Omega^{-1}$.

of $|n| = 1$ (Eq. (4)) without sine terms. Primarily for this reason, nodal knots have not appeared in existing linearized reciprocal circuit architectures, or related settings of classical topological matter.

In this work, our key insight is to instead realize pairs of nodal knots related by mirror symmetry, such that reciprocity does not have to be broken. This can be achieved via a mapping $F(\mathbf{k}) = (z(\mathbf{k}), w(\mathbf{k}))$ such as

$$z = \cos 2k_z + \frac{1}{2} + i(\cos k_x + \cos k_y + \cos k_z - 2)$$
$$w = \sin k_x + i \sin k_y,$$
(7)

which possesses opposite windings of $n \approx \pm 1$ in each of the two halves of the 3D BZ given by $k_z > 0$ and $k_z < 0$ (Fig. 1b). Provided that $w$ is raised only to even powers in $\bar{f}(z, w)$, the Laplacian will be even in $\mathbf{k}$, and hence realizable in an RLC, and as such reciprocal, circuit.

The overwhelming advantage of topolectrical circuit array implementations is that nodal structures naturally manifest as robust impedance peaks, i.e., electrical resonances. Consider a multi-terminal measurement with input currents and potentials given by the $I_\alpha$ and $V_\beta$ components respectively (c.f. Eq. (6)). In general, the impedance $Z_{ab}$ between modes $a$ and $b$ is given by

$$Z_{ab} = \sum_\lambda \frac{|\psi_\lambda(a) - \psi_\lambda(b)|^2}{j_\lambda},$$
(8)

where $j_\lambda$ and $\psi_\lambda$ are the corresponding eigenvalues and eigenvectors of the circuit Laplacian $J$. Note that the modes $a$, $b$ are not necessarily the real-space nodes $\alpha$, $\beta$ appearing in Eq. (6); in the translation-invariant circuits that we consider, they can also refer to quasi-momentum modes from the Fourier decomposition of multiterminal measurements. Importantly, for circuits designed such that $j_\lambda \approx 0$ along the nodal loops/knots or their drumhead regions, $Z_{ab}$ should signal pronounced divergences (resonances) when either $a$ or $b$ coincide with the nodal regions. More generally, $Z_{ab}$ should diverge strongly whenever the Laplacian exhibits a zero-eigenvalue flat band with divergent density of states, since $j_\lambda \approx 0$ for extensively many $\lambda$, unless $\psi_\lambda(a) = \psi_\lambda(b)$ at terminal $a$, $b$.

For the sake of concreteness, we specialize to a periodic circuit network with a repeated unit cell structure. This allows us to rewrite Eq. (6) as

$$I_{(\mathbf{x},i)} = J_{(\mathbf{x},i),(\mathbf{y},j)} V_{(\mathbf{y},j)},$$
(9)

with $\mathbf{x}$, $\mathbf{y}$ labeling the unit cell positions in the circuit, while $i, j = \{1, 2\}$ labels the two sublattice nodes inside each unit cell. By

exploiting the translational invariance of unit cells in the circuit, $J_{(\mathbf{x}, i),(\mathbf{y}, j)} = J_{i,j}(\mathbf{x} - \mathbf{y})$, we can find the irreducible representations of the translational group of $J$ by a Fourier transformation in the real space coordinates

$$J_{i,j}(\mathbf{k}) = \sum_{\mathbf{r}} J_{i,j}(\mathbf{r}) \, e^{-i\mathbf{k}\cdot\mathbf{r}}.$$
(10)

In Eq. (10), we sum over all unit cell positions $\mathbf{r}$ in the circuit network. We define the Fourier transformation of $J$ to be in the directions perpendicular to the open boundary surface. The dimension of the resulting matrix $J(\mathbf{k})$ is fixed by the number of circuit nodes that do not transform into each other by translation. By diagonalizing $J(\mathbf{k})$, we find the admittance band structure $j_n(\mathbf{k}), n \in \{1, \ldots, \dim(J(\mathbf{k}))\}$ of the circuit network as a mapping of quasi-momentum $\mathbf{k}$ to admittance eigenvalues of $J$. The fully periodic circuit network is then constructed such that the admittance band eigenvalues are given by the absolute value of $f$, $j_\pm(\mathbf{k}) = \pm|f(\mathbf{k})|$. The kernel of the fully periodic admittance band structure features one-dimensional closed nodal loops in its 3D BZ, that are induced by the corresponding mapping $\mathbb{T}^3 \to \mathbb{C}$ inherited from the function $f(\mathbf{k})$. In an experimental setting, it is possible to extract the admittance band structure by performing $N$ linearly independent measurement steps, where $N$ describes the number of inequivalent nodes in the network. Each step consists of a local excitation of the circuit network and a global measurement of the voltage response, from which all components of the Laplacian in reciprocal space can be extracted. Consequently, the admittance band structure is found by a diagonalization of $J(\mathbf{k})$ for each $\mathbf{k}$.

In the following, we show $Xyce$[41] simulation results of the prescribed measurement procedure with periodic (Fig. 3) as well as open boundary conditions (Fig. 4) for circuits featuring a Hopf-link, trefoil knot and figure-8 knot. The experimental details for the Hopf-link are described in the Methods section.

Before proceeding to more involved nodal knots, we illustrate our approach through the simplest example of a nontrivial linked nodal structure—the Hopf-link (Fig. 1c). With $f(\mathbf{k}) = z(\mathbf{k})^2 - w(\mathbf{k})^2$ ($z_{1,2}(s) = \pm e^{is}$ in Eq. (3)), it is the simplest possible nontrivial nodal structure, with at most next-nearest neighbor (NNN) unit cells connected by capacitors $C$, $C/2$, $C/4$ or inductors $L$, $L/2$, $L/4$ in each direction (see "Methods"). In steady-state $Xyce$ AC simulations, where the frequency parameter is set by the external excitation, the impedance peaks at $\omega^2 = \frac{1}{LC}$ indeed accurately delineate the two inter-linked nodal rings, as shown in Fig. 3a. Its surface projections are even more accurately resolved as drumhead regions when the measurements are taken on open

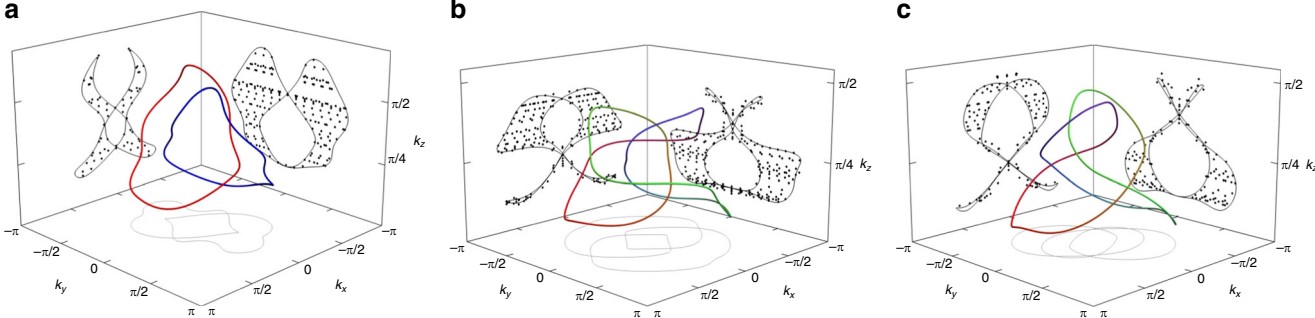

**Fig. 4 Simulated drumhead state measurements under various OBCs.** The black diamond-shaped points indicate points in reciprocal space with admittance eigenvalues smaller than their respectively admittance thresholds $j_x$, $j_y$ corresponding to $x$, $y$ open boundaries, as obtained from *Xyce* circuit simulations. These points are contained in regions of the surface BZ which are bounded by the projected 3D theoretically computed bulk nodal structures (colored green, red and blue). **a** shows two entangled unknots, defined as the Hopf-link. The black dots combine the simulation results with circuit dimensions of ($22 \times 22 \times 16$), ($23 \times 23 \times 20$), ($16 \times 22 \times 19$), ($22 \times 22 \times 14$), ($25 \times 20 \times 23$) and ($25 \times 24 \times 23$). The admittance thresholds are chosen to be $j_x = 0.0027\,\Omega^{-1}$ and $j_y = 0.0020\,\Omega^{-1}$. **b** depicts a trefoil knot showing the combined simulations of circuit system sizes of ($20 \times 20 \times 20$), ($21 \times 21 \times 21$), ($24 \times 15 \times 15$), ($21 \times 20 \times 25$), ($18 \times 19 \times 17$), ($17 \times 18 \times 21$), ($23 \times 21 \times 19$), ($19 \times 25 \times 23$) and ($20 \times 20 \times 22$). The admittance thresholds are chosen to be $j_x = 0.0030\,\Omega^{-1}$ and $j_y = 0.0025\,\Omega^{-1}$. **c** illustrates a figure-8 knot with ($23 \times 23 \times 23$), ($20 \times 20 \times 25$), ($20 \times 20 \times 21$), ($19 \times 16 \times 18$), ($17 \times 14 \times 16$), ($19 \times 25 \times 25$) and ($25 \times 21 \times 22$) unit cells in the respective directions. The admittance thresholds are chosen to be $j_x = 0.0028\,\Omega^{-1}$ and $j_y = 0.0032\,\Omega^{-1}$.

boundary surfaces normal to $\hat{x}$ and $\hat{y}$, as shown in Fig. 4a. No drumheads are expected for $\hat{z}$ open boundary surfaces, since there is another mirror-image nodal structure related by $k_z \rightarrow -k_z$.

We next consider the trefoil knot, which is defined by $f(\mathbf{k}) = z(\mathbf{k})^2 - w(\mathbf{k})^3$. While it, even after topology-preserving real-space truncations (see "Methods"), still necessitates longer-ranged connections, circuit networks conveniently allow to accomodate for such couplings. In Figs. 3b and 4b, we present the simulation results of the detailed imaging of a nontrivially knotted nodal loop and its drumhead surface projections, which also showed remarkable agreement with theoretical expectations.

Our approach can also be conveniently applied to more obscure non-torus knots where $f(z, w)$ is not a polynomial in $z$ and $w$. For illustration, we simulate the circuit with a Figure-8 knot nodal structure with $f(\mathbf{k}) = 64\,z(\mathbf{k})^3 - 12\,z(\mathbf{k})(3 + 2(w(\mathbf{k})^2 - \bar{w}(\mathbf{k})^2)) - 14(w(\mathbf{k})^2 + \bar{w}(\mathbf{k})^2) - (w(\mathbf{k})^4 - \bar{w}(\mathbf{k})^4)$, where $w(\mathbf{k})$, $\bar{w}(\mathbf{k}) = \sin k_x \pm i \sin k_y$. The Figure-8 knot belongs to the more general class of knots known as lemniscate knots, where the equivalent braid cannot be expressed the braiding of $p$ strands with $q$ revolutions, and requires the appearance of both $w$ and $\bar{w}$ in its $f(\mathbf{k})$[35]. Despite its ostensibly more complicated appearance, its nodal structure and surface drumhead states, shown in Figs. 3c and 4c, respectively, can be easily obtained from impedance measurements.

**Experimental mapping of surface drumhead states.** A highlight of this work is the experimental verification of our design of momentum-space nodal structures. Due to the topological significance of surface drumhead states, as well as their extensively large density of states, our experiment shall involve the mapping of the drumhead state of the nodal Hopf Link shown in Fig. 4a, where $k_y$ and $k_z$ are synthetic coordinates. This surface was chosen due to the distinctive "double-loped" structure of the drumhead state, which should prominently show up as a region of elevated topolectrical impedance.

The first step in experimental circuit design is to simplify the real-space lattice structure. After optimal truncation and tuning of the x-direction couplings (see "Methods"), we obtained a slightly modified Hopf-link with qualitatively similar double lobes in its drumhead region (Fig. 5a). Note that unlike the topological drumhead modes themselves, the elevated region consists of extra "ridges and valleys" due to additional contributions from other bands in Eq. (8). This circuit is physically implemented with an array of connected printed circuit boards (PCBs), each

representing one unit cell, which can be adjusted to accurately correspond to different $(k_y, k_z)$ points by tuning the inductors (Fig. 6 of Methods). Enabled by individually addressing the nodes, our tuning approach allows each inductance to be reliably adjusted by $-50\%$ to $+25\%$ of its original manufactured value, realizing to our knowledge the most accurately tunable circuit in the literature of topolectrical circuits to this date. To realize the required variety of capacitance values, we have implemented each logical capacitor as an appropriate parallel configuration of a few commercially available capacitors (see "Methods"). All parametric tunings are relegated to the inductances, since variable inductors are more reliably tuned than variable capacitors in practice.

While the topological robustness of drumhead states increases with the number of unit cells $N$, so do the destabilizing contributions from parasitic resistances and components uncertainties. As simulated in Fig. 5b for realistic component values, we have found that a rather low $N = 9$ already gives rise to a robustly visible drumhead region of elevated impedance. Importantly, this robustness is well corroborated against the experimental impedance data presented in Fig. 5c. Even with only 14 $(k_y, k_z)$ data points, each obtained through careful tuning, we have observed a very high fidelity between the expected and measured impedance values, as also visually evident from the almost perfect match of the blue/red (low/high imepdance) points between simulation and experiment (Fig. 7 of Methods). To mitigate the effects of parasitic resistance and component uncertainty, we have also taken advantage of a machine learning algorithm that choses $(k_y, k_z)$ sampling points that remain the most impervious to these uncertainties (Fig. 8 of Methods).

Besides conclusively demonstrating the experimental viability of mapping out nodal drumhead states, our experiment also pushes the state-of-the-art in tunable topolectrical circuits, where even minute unevenness between unit cells can potentially affect the circuit band structure significantly. As further elaborated in the "Methods" section, further refinement of this technique through micro-controllers can lead to even more accurate automated tuning that can eventually realize topological pumping in quasiperiodic (Aubry-Andre-Harper) circuits.

## Discussion

We have introduced an experimentally accessible approach for realizing generic momentum space nodal knots. Our proposed systems can be easily implemented in RLC circuit setups, whose

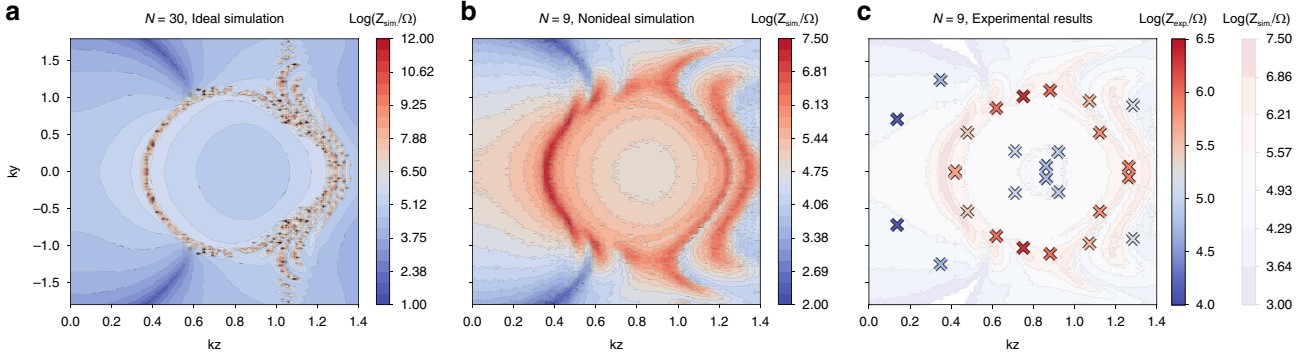

**Fig. 5 Simulated impedances vs. experimental measurements. a** Hopf-link (dark cyan) and the drumhead region (orange) of elevated impedance it encloses, computed in the "clean" limit absent of parasitic resistances and component uncertainty. Simulation was performed with $N = 30$ unit cells at resonant frequency 795.7 kHz for the circuit Laplacian in Eqs. (15) to (18) (in "Methods"), truncated from that of Fig. 4a to facilitate experimental construction. **b** Impedance map of the same circuit, but simulated for our $N = 9$ experimental setup with empirically determined parasitic inductor and capacitor resistances $R_{pL} = 0.11\,\Omega$ and $R_{pC} = 0.03\,\Omega$, and capacitor/inductor tolerances of 1%. **c** Corresponding experimentally measured impedance (crosses) with distinct elevated region, which agree well with simulation (lighter background contours from (**b**)). Frequency used is 740 kHz, offset from the predicted 795.7 kHz to account for uncertainties in the tuning circuitry (see "Methods" and Supplementary Table 3).

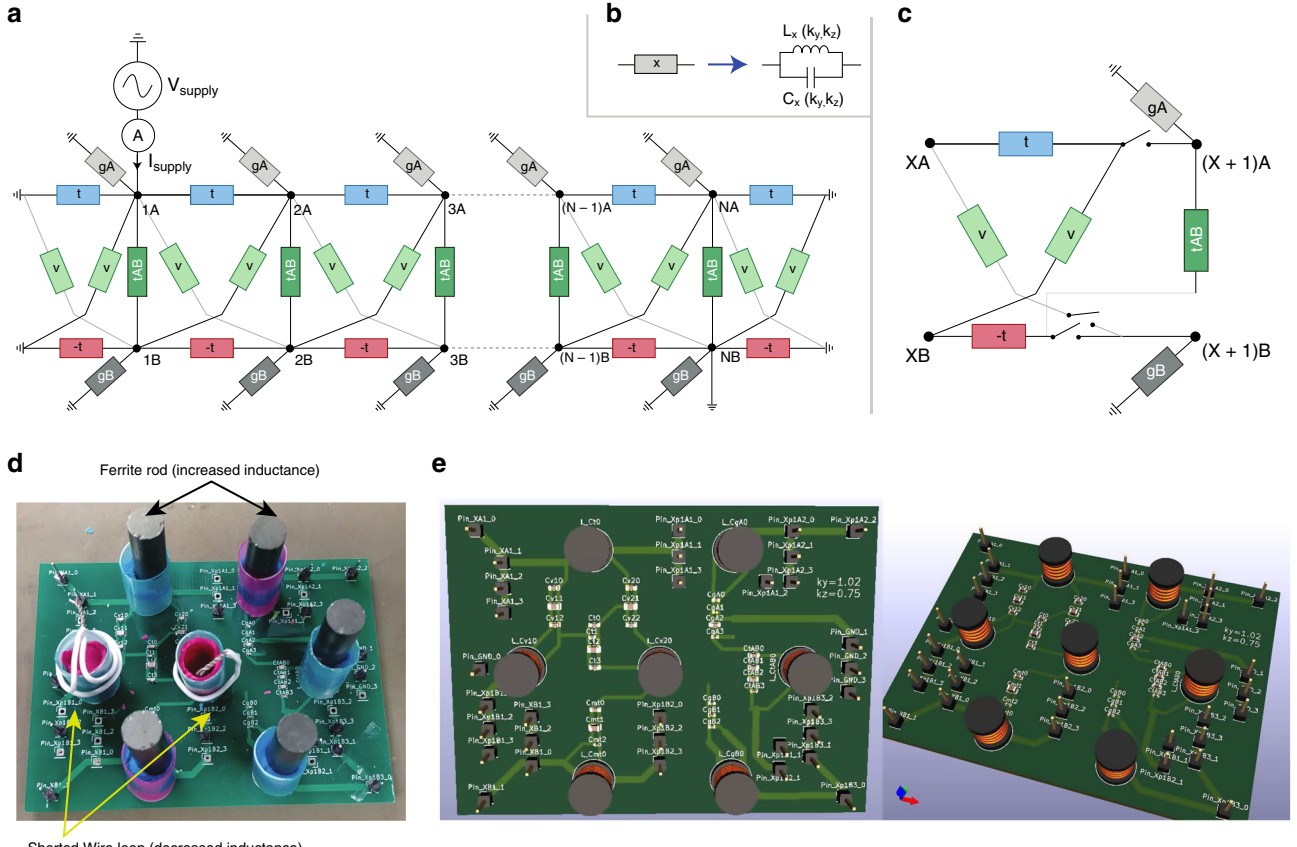

**Fig. 6 Schematic and PCB implementation of Hopf-Link experiment. a** Schematic of our 2-leg ladder LC circuit array, whose Laplacian takes the form of the Hopf-link at resonance when the component admittances are chosen according to (15)–(18). **b** Each rectangle in (**a**) corresponds to a parallel combination of an inductor and a logical capacitor whose specifications are indicated in Supplementary Table 1. As explained in the main text, each inductor can be accurately tuned to vary $k_y$, $k_z$ near the drumhead region. **c** Schematic representation of one repeating unit cell of the circuit used to construct final experiment. The switches are set to open when the inductors are being tuned, and closed when the impedance of the entire circuit is measured to map out the drumhead region. **d** Experimental PCB realization of one repeating unit. Visible are the inductors equipped with ferrite rods or shorted wire loops, which respectively increase/decrease the inductances in a tunable manner. **e** Renderings of the same PCB to emphasize its physical structure. Each large cylinder represents a variable inductor, while the components prefixed by "*C*" represent capacitors that are connected in parallel to form the logical capacitors in Fig. 9. Detailed specifications of these components are given in Supplementary Tables 1 and 2.

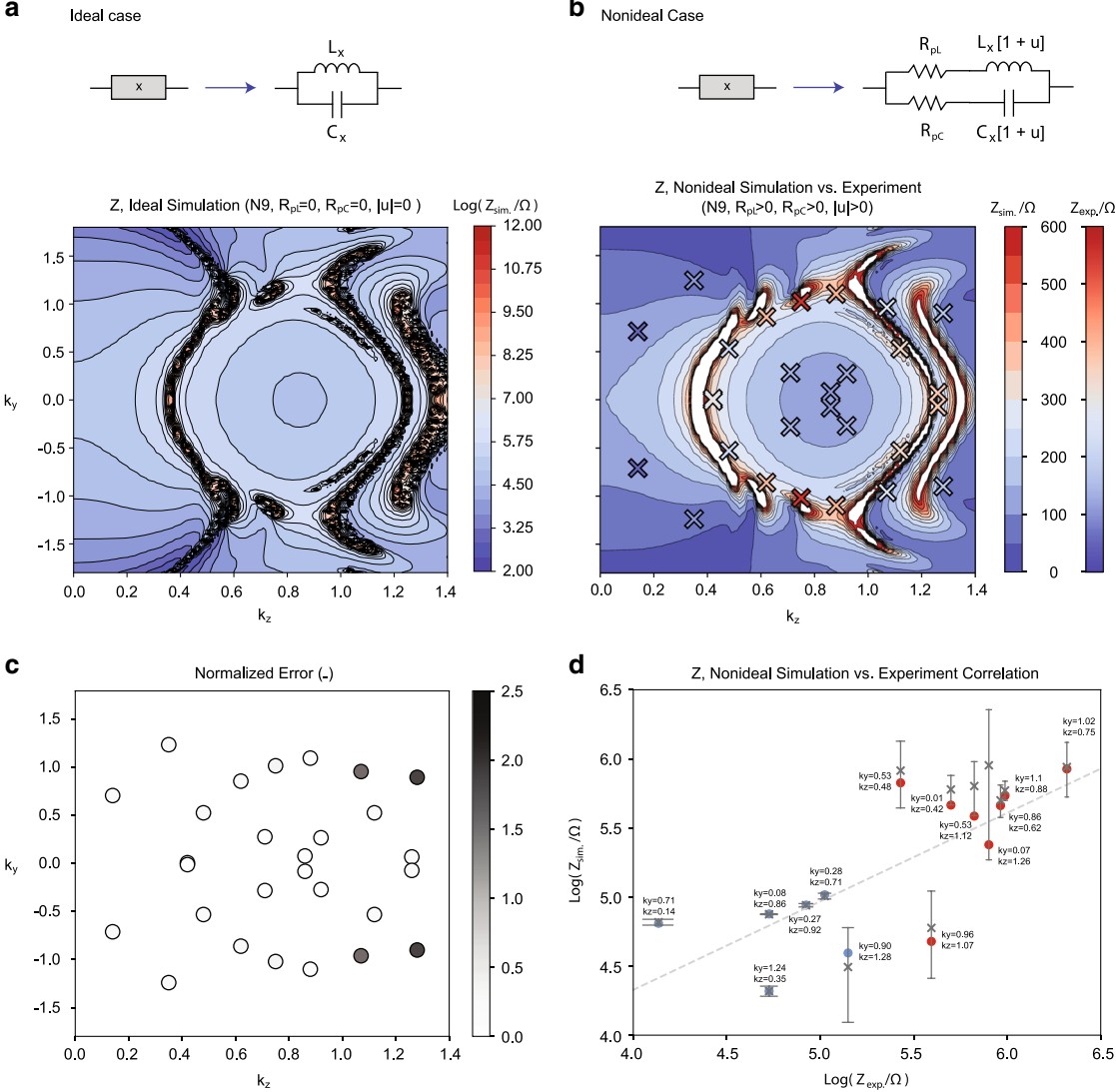

**Fig. 7 Experimental vs. simulations with ideal/nonideal components. a** Simulated impedance map for $N = 9$ with ideal components specified by Supplementary Table 1 with no parasitic resistance or uncertainty, with an elevated drumhead region clearly visible. White regions denote impedance values above 600 Ω. **b** Contour plot of simulated impedance map for the same scenario as in (**a**), but with parasitic resistance $R_{pL} = 0.11$ Ω, $R_{pC} = 0.03$ Ω and random variation in inductor and capacitor values $u \in [-0.01, 0.01]$, $Z_{sim.}$, and Experimentally measured impedance $Z_{exp.}$ (colored crosses), see Supplementary Table 3. **b** Normalized error of measured points, Normalized Error $= (|Z_{exp.} - Z_{sim.}|/Z_{sim.})^2$. **c** Plot of the Log of expected simulated impedance, Log($Z_{sim.}$), vs. the Log of experimentally measured impedance, Log($Z_{exp.}$) (blue and red dots). The blue dots represent points in the low-lying regions, while the red dots are located near the drumhead region. The gray dashed line is the line computed by least squares regression. Gray crosses represent the mean of the simulated impedance within a 0.03 radius in k space surrounding a particular $k_y$, $k_z$ point, while error bars represent the standard deviation of impedance within the 0.03 k radius of that particular $k_y$, $k_z$ point, which is shown in Supplementary Tables 4–14. For $k_y$, $k_z$ points in regions with very high local standard deviation, the gray cross may not coincide with the blue/red dots. The correlation coefficient between Log($Z_{sim}$) and Log($Z_{exp.}$) is 0.743, and increases to 0.863 when the three borderline points with largest variance are excluded.

nodal admittance band structure is directly characterizable via impedance measurements. A key theoretical novelty for accomplishing this is our choice of momentum space embedding functions $z(\mathbf{k})$, $w(\mathbf{k})$, which permits the knotting (and not just linking) of momentum space nodal structures without breaking reciprocity. This not only allows for easy implementation of almost any desired knot from its corresponding braid, but also for a robust surface drumhead state characterization of the knots. Combined with multi-terminal impedance measurements in the bulk, our RLC nodal knot framework provides an unprecedentedly direct access to the Seifert surface structure and knot invariants. Our approach is explicitly demonstrated through large-scale simulations of three different nodal knot circuits, as

well as an experiment which maps out the drumhead surface state of a nodal Hopf-link. It established the proof of principel how to realize any nodal knot in a topolectric circuit.

As the next refinement step of the analytic simulation of the electronic setup, one needs to take into account parasitic resistances, in particular those that derive from the inductors. Here, the dissipative, i.e., non-Hermitian generalization of our idealized Hermitian circuit setup opens up yet another unexplored territory of topological matter[42–44], i.e., non-Hermitian nodal knot systems[40,45]. We defer this analysis to future work. In order to directly remedy the parasitic effect from the inductors, the most viable solution is to increase the AC frequency scale into the Megahertz regime at which the nodal knots are observed. This

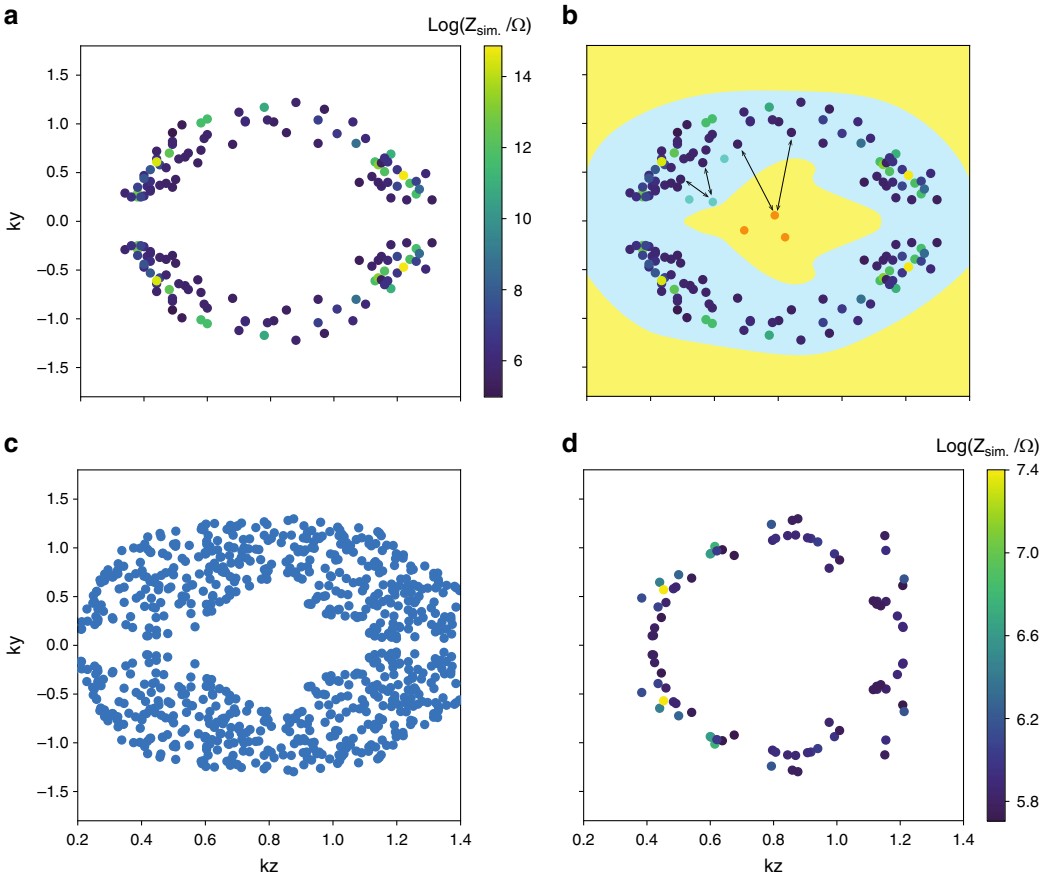

**Fig. 8 Machine learning optimization of ($k_y$, $k_z$) measurement points. a** Initial suboptimal set of ($k_y$, $k_z$) sampling points for the drumhead region, subject to a relatively relaxed criterion of $\log |Z_{avg} - Z_{SD}| > 5.2$, where $Z_{SD}$ is the standard deviation of the impedance subject to 1% tolerance in the capacitances and inductances with parasitic resistance $R_{pL} = 0.11\,\Omega$, $R_{pC} = 0.03\,\Omega$. While possessing higher impedance than points outside the drumhead region with $|Z_{avg}| < 4.8$, these still suffer from significant uncertainty effects (motley of colors). **b** The Nearest-Neighbor algorithm sets an allowed region (light blue) for new ($k_y$, $k_z$) points, which are at most a distance 0.1 away from at least two existing good sampling points. **c** New randomly generated unfiltered sampling points in the allowed region. **d** Output consisting of new sampling points filtered according to more stringent criteria $\log |Z_{ideal}| > 5.7$, $|(Z_{avg} - Z_{ideal})/Z_{ideal}| < 0.2$, $Z_{SD}/Z_{ideal} < 0.2$, which only need to be sieved out from the allowed region.

would also help with the higher spatial intergration or our nodal knot circuits. Setting up a new generation of Meghertz topolectric circuits will hence be a prioritized experimental future objective.

## Methods

**Circuit simulation details**. This section elaborates on the setup of the circuits that we simulated. As detailed in the main text, the desired knot or link is given by the kernel of a knot function $f(z, w)$ that maps the 3D BZ $\mathbb{T}^3$ to a complex number $\mathbb{C}$. The first step in determining the circuit design is the construction of the function $f(z, w)$ from the corresponding braid through the procedure we had outlined. In the next step, we find suitable functions $z(\mathbf{k})$ and $w(\mathbf{k})$ that faithfully map the knot to the kernel of $f(\mathbf{k})$. To be able to implement the corresponding function $f(\mathbf{k})$ in a circuit environment i.e. a tight-binding lattice that preserves reciprocity, we implement two mirror images of the circuit in the BZ that are related by $k_z \to -k_z$. The Laplacian for the circuit simulations is then set up as (note the slightly different definition of $f$ from Eq. 1 of the main text)

$$J(k_x, k_y, k_z) = i\omega_0 C \left[ \Im m f(k_x, k_y, k_z)\, \tau_x + \Re e f(k_x, k_y, k_z)\, \tau_z \right]. \quad (11)$$

The circuit connections are then designed such that they form the Laplacian $J(\mathbf{k})$. This is achieved by expanding the real and imaginary part of $f$ as single cosine terms and implementing the separated terms as internodal connections in the circuit. Those connections need to fulfill two criteria. First, they need to realize the proper real space linkage between two nodes to replicate the specified term in the ($2 \times 2$) Fourier transformed Laplacian. Second, the magnitude of those elements is to scale with the prefactor of the corresponding cosine term. A positive value is implemented by a capacitor and a negative value by an inductor. Finally, we need to account for the total node conductance in the circuit setup by implementing adequate grounding terms. The scales of the capacitances and inductances are

chosen to be $C = 1$ nF and $L = 10$ μH, yielding a resonance frequency of

$$f_0 = \frac{1}{2\pi\sqrt{LC}} \approx 1.592 \text{ MHz}. \quad (12)$$

$f_0$ will be the operating frequency for all performed simulations, where signatures of the prescribed nodal knots or links emerge. At this specific frequency, the inductances defined act as negative capacitances due to their $\pi$ relative phase shifts. For reasons of numerical stability, we include additional ground connections of $C_{ground} = 100$ nF and $R_{ground} = 1$ kΩ at every node in the circuit. These terms just enter as an identity matrix contribution $l_0\mathbb{I}$ and can be subtracted out after the band structure has been reconstructed from the simulation data. The Laplacian of the circuit is then shifted as $J(\mathbf{k}) \to J(\mathbf{k}) + l_0\mathbb{I}$, and its two band admittance spectrum is given by

$$\begin{aligned} j_{\pm}(k_x, k_y, k_z) &= l_0 \pm i\omega_0 C \sqrt{(\Re e f)^2 + (\Im m f)^2} \\ &= l_0 \pm i\omega_0 C\, |f|. \end{aligned} \quad (13)$$

To recreate the admittance band structure, we use the measurement scheme initially described in[19]. There and in all our simulations, each measurement step consists of a local excitation of the circuit at one node through an AC driving voltage via a shunt resistance and a global measurement of the total voltage profile at all nodes in the circuit. The shunt resistance enables the measurement of the input current that is fed into the circuit.

From the global response of the circuit, we can reconstruct the Fourier coefficients of $J$ in reciprocal space and diagonalize $J(\mathbf{k})$ for every $\mathbf{k}$. This measurement procedure must be repeated $M$ times, where $M$ describes the number of non-equivalent nodes in the circuit network to be able to reconstruct the full Laplacian $J(\mathbf{k})$. From the admittance band structure, we then distill the closed nodal loops of the specified model by selecting the imaginary admittance eigenvalues, that are smaller than a globally chosen upper threshold. This upper bound is selected such that the valley points corresponding to the zero nodal points on the knot or

link are recovered, but no additional points appear in regions with small gradients close to the nodal line. Due to the discretization of the BZ, we recover only a discrete set of nodal points in the BZ. This drawback can be counterbalanced to some degree by simulating circuit networks with different dimensions in terms of unit cells. This way, we enhance our grid resolution in reciprocal space and obtain a more precise result due to an increased number of data points on the knot or link.

Similarly, the OBC simulations are evaluated by extracting admittance eigenvalues smaller than a chosen limit. Those points in the projected BZ form 2D areas, as shown in Supplementary Fig. 2. These 2D areas correspond to projections of the Seifert surface bounded by the corresponding link or knot onto the direction of the open boundary surface. The corresponding zero-admittance eigenstates amount to the so-called Drumhead states that are exponentially localized at the boundary with an inverse localization lengths given by their imaginary gaps[46,47]. With these preliminary explanations, the only remaining requisite to perform the individual simulations is the specification of the employed knot function $f(z, w)$ and the functions $z(\mathbf{k})$ and $w(\mathbf{k})$. Note that since $f(\mathbf{k})$ in general consists of an exponential tail of distant couplings in real space[46,48,49], some gap-preserving real space truncation of its real and imaginary parts is necessary for actual implementations. For the most part, this presents no additional challenges, and can be adapted to conform to the specifications of available actual electronic components. We also need to define an upper admittance threshold for resonance to extract the nodal points from the obtained simulation data.

We perform *Xyce* simulation for different system sizes in order to increase the resolution of the knot in the BZ. Since the reciprocal space consists of discrete points of allowed quasi-momenta for any finite number of unit cells, we cannot to trace out the knot exactly. In order to increase the density of samples, one can increase the system size, but this increases the computational costs. Our alternative approach is to create several copies of the same setup, but with varying system sizes. Choosing the number of unit cells as co-primes of one another increases the sampling density of the combined momentum grid without the need for creating a very large system.

In the Xyce simulations, we create *spice* netlists which represent a circuit network consisting of capacitors and inductors described by a Laplacian in the form of (11) and perform AC analyses on them. In order to simulate one measurement procedure step necessary to reconstruct the admittance band structure, we connect an ideal voltage source via a shunt resistor to the circuit. As the parameters of amplitude voltage and shunt resistance can be chosen arbitrarily in a simulation, we used 1 V and 1 Ω. The AC analysis frequency is given by $f_0$.

**Drumhead state experiment**. The objective of our experiment is to reconstruct the surface topological drumhead state of a simplest illustrative nodal structure, the Hopf-link, as shown in Supplementary Fig. 2a. For that, the physical circuit must possess a Laplacian $L^C$ that is proportional to the Laplacian $L$ of a Hopf-link at a particular resonant AC frequency $\omega_0$. For a streamlined implementation, we deformed the Hopf-link Laplacian from the Supplementary Information such that it contains up to only nearest neighbor (NN) connections along the surface normal $\hat{x}$ while retaining a qualitatively similar nodal structure (Fig. 5). Explicitly, we require

$$L^C\big|_{\omega=\omega_0} = i\omega\big(L_z^C L_x^C L_x^C - L_z^C\big)\big|_{\omega=\omega_0} \propto i\omega_0\big(L_z L_x L_x - L_z\big), \quad (14)$$

where the components of the deformed Hopf-link Laplacian are given by $L_z = 4\cos k_x(2 - \cos k_y - \cos k_z) - 2(5 - 4\cos k_y + \cos 2k_y + \cos k_y - k_z - 4\cos k_z - \cos 4k_z + \cos k_y + k_z)$ and $L_x = (1 + 2\cos 2k_z)(\cos k_z + \cos k_y + \cos k_x - 2)$. One way to satisfy Eq. (14) is to design the physical circuit such that its corresponding components $L_x^C, L_z^C$ are of the forms

$$L_x^C = -t_{AB} - 2v\cos k_x, \quad (15)$$

$$L_z^C = 2t(1 - \cos k_x) + g_A + t_{AB} + 2v, \quad (16)$$

where $v$, $t$, $t_{AB}$, $g_A$ and $g_B$ depend parametrically on $k_y$, $k_z$ as follows:

$$
\begin{aligned}
v &= 1 + 2\cos 2k_z \\
t &= 4(2 - \cos k_y - \cos k_z) \\
t_{AB} &= 2(\cos k_y + \cos k_z - 2)(1 + 2\cos 2k_z) \\
g_A &= 6 + 4\cos 2k_y + 4\cos k_y - k_z - 12\cos k_z \\
&\quad + 4\cos 2k_z - 2(5 + 2\cos 2k_z)\cos k_y - 2\cos 3k_z \\
&\quad - 4\cos 4k_z + 4\cos k_y + k_z \\
g_B &= -2 - 4\cos 2k_y - 4\cos k_y - k_z + 4\cos k_z \\
&\quad + 4\cos 2k_z + 2(3 - 2\cos 2k_z)\cos k_y \\
&\quad - 2\cos 3k_z + 4\cos 4k_z - 4\cos k_y + k_z
\end{aligned}
\quad (17)
$$

() can be realized with an LC circuit array in the form of a 2-leg ladder with $N$ unit cells (rungs) and $2N$ nodes in total (Fig. 9). Each term $x \in [t, v, -t, t_{AB}, g_A, g_B]$ is represented by a parallel configuration of a tunable inductor $L_x$ and capacitor $C_x$

of appropriate value, such that its admittance

$$G_x(k_y, k_z) = i\omega C_x + \frac{1}{i\omega L_x} = i\omega C_0\left(c_x - \frac{1}{\omega^2 L_x C_0}\right) \quad (18)$$

is of the required $(k_y, k_z)$-dependent value $t$, $v$, $-t$, $t_{AB}$, $g_A$ or $g_B$ at a particular $\omega = \omega_0$. As elaborated later, it suffices to vary only the inductances to sweep through the entire range of $(k_y, k_z)$ stipulated by the size of the drumhead region in Fig. 5. Here $C_0$ is an arbitrarily defined reference capacitance value that offers a free rescaling degree of freedom in the tuning, and $c_x$ is the corresponding dimensionless capacitance of element $x$. Each element proportional to $2(1 - \cos k_x)$ couples two neighboring unit cells, with each term in the off-diagonal $L_x^C$ couples the upper and lower rungs. Note that our proposed circuit requires only LC components i.e. inductors and capacitors, with positive and negative resistors truncated off without appreciably changing the shape of the drumhead region. That said, with the contact and parasitic resistances intrinsic to an experimental circuit, some of these resistances will be inevitably reintroduced. These, however, also lead to no significant modification of the drumhead region, as verified via a simulation with realistic amounts of parasitic resistances and component uncertainty (Fig. 7b).

Our circuit is built with interconnected PCBs, each representing one unit cell, as shown in Fig. 9c. With a strategic choice of $C_0$ and frequency $\omega_0$, it is possible to scan through the entire relevant range of $k_y$, $k_z$ by just tuning the inductances alone. As elaborated later, this can be accurately achieved through the use of ferrite rods and shorted wire loops within/around each inductor. The required fixed capacitances are realized by parallel combining commercially available capacitances into logical capacitors. The specifications of these logical components, as well as that of their underlying physical capacitors, are detailed in Supplementary Tables 1 and 2.

A major consideration of topolectrical circuit design is that imperfections from parasitic/contact resistances and component uncertainties should not change the measured impedance and hence Laplacian band structure significantly. Inductors are commonly manufactured with +/−10% inductor value uncertainty, and a typical parasitic resistance that scales at a rate of 2.45 Ω/1 mH. In theory it is possible to decrease the impact of parasitic resistance by increasing the inductance, but this cannot be done in practice because larger inductors typically require longer wires which increases parasitic resistance. Capacitors on the other hand are commonly manufactured with +/−5% uncertainty, and have negligible parasitic resistance compared to PCB trace wires, which possess 0.024 Ω/cm. In the case of capacitors, the effect of parasitic resistance may be decreased by picking smaller capacitors. However, choosing smaller capacitors requires larger inductors for the same frequency used in impedance measurement, which increases parasitic resistance, or requires a higher frequency. Therefore ideally one chooses the $\omega_0$ to be as high as possible depending on their signal generator/impedance measurement equipment, and then chooses a value of $C_0$ that results in the smallest effect of parasitic resistance in the capacitors, but not too small to increase the inductor values and inductor parasitic resistances. Such imperfections can be modeled as additional serial resistivities on inductances and capacitances that are rescaled by a factor of $1 + u$, $u$ a random variable, as illustrated in Fig. 9b for the measured impedance across the entire circuit (between nodes 1A and $N$B).

Impedance data measurement and analysis: To map out the drumhead state, we measured the impedance across the first and last nodes of the circuit (1A and $N$B) at a number of strategically determined $(k_y, k_z)$ points that are relatively insensitive to component disorder, as elaborated in the following subsection. As presented in Fig. 5, the drumhead region is indeed clearly visible as a region of elevated impedance, in close agreement with the imperfection-corrected simulation (Fig. 7b.). The simulation also revealed that parasitic resistance in general decreased the impedance contrast by reducing the high impedance in the drumhead region and raising the low impedance outside of it. Component uncertainty increases the variance in the measured impedance at each individual $(k_y, k_z)$ point, such that a larger number of measured $(k_y, k_z)$ points are needed to average over the noise.

While a very large $N$ will yield the most topologically robust drumhead state in an ideal setting, in practice that will also introduce much larger accumulated parasitic resistances and total component uncertainties, not to mention the copious resources needed. As such, we have built our circuit with $N = 9$ cells as a compromise between topological localization, noise and cost. The complete setup is pictured in Fig. 10. After tuning all inductors in accordance to the $k_y$, $k_z$ values, the impedance across the entire circuit is measured by attaching nodes 1A and $N$B to a voltage divider and observing the voltage drop across the circuit using an oscilloscope. After correcting for possible frequency shifts due to uncertainties in the tuning circuitry (elaborated later), we indeed measured a distinctive cluster of elevated impedances in the drumhead region, as shown in Fig. 5c and analyzed in Fig. 7.

Even though the experimental setup also suffers from imperfect tuning of inductor values and additional parasitic resistances from the solders linking the repeating PCBs (unit cells), we are still able to reliably distinguish the low/high impedance points and hence delineate the correct drumhead region. Experimentally, we were able to measure 5 points in the low-lying regions $Z_{\text{exp.}} < 160\Omega$, 6 points in the elevated region $Z_{\text{exp.}} > 250\Omega$ and 3 points in the borderline region between them. The 5 points in the low-lying region correspond to the "inside" of the elevated region $k_y < 0.5$, $0.6 < k_z < 1.0$, and the region to the left of the elevated region $k_z < 0.4$. The square root of the average normalized error

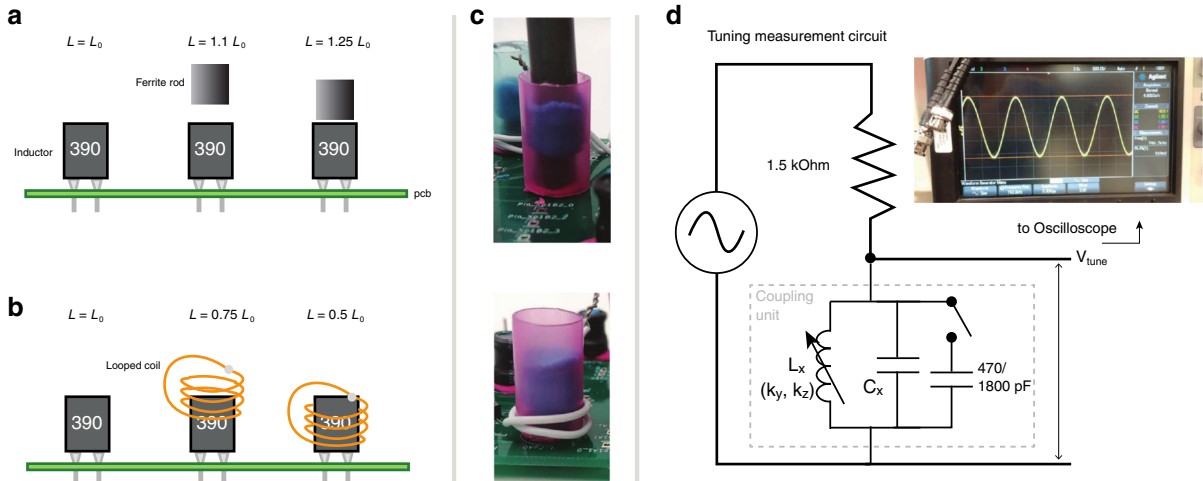

**Fig. 9 Tuning of variable inductors.** Illustration of tuning methods of the variable inductors and circuit diagram of impedance measurement circuit for tuning inductors and imaging the drumhead region. **a** By adding a ferrite rod to the top of a fixed-value inductor, the inductance may be increased by up to 25%. **b** By surrounding the inductor with a shorted wire loop, the inductance may be decreased by up to 50%. **c** Experimental implementation of ferrite rod and wire loop. A plastic straw and modeling clay is used to hold the ferrite rod/wire loop in place after tuning. **d** External circuit used to measure and tune the impedance of each coupling unit (logical component), as described by (19).

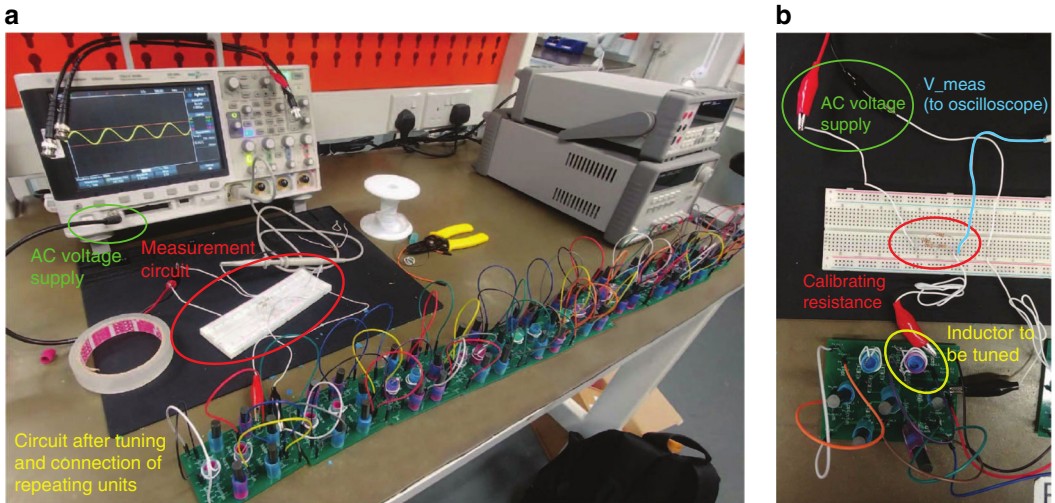

**Fig. 10 Experimental setup for impedance measurement. a** Full view of setup for impedance measurement of the complete circuit array. The oscilloscope generates the AC voltage signal for the impedance measurement, as well as for tuning each variable inductor of each coupling unit (logical component). Schematics of the circuits are described in Figs. 9 and 6. **b** Close up of measurement circuit with AC voltage supply, test resistor forming a voltage divider, inductor currently being tuned, and wire connecting to oscilloscope to measure voltage across the coupling unit.

of each measured point was 0.2, and the coefficient of correlation between simulated and measured data were 0.743. When excluding three points $(k_y, k_z)$ = (0.9, 1.28), (0.96, 1.07), (0.07, 1.26), which according to simulation are in regions with very high local variance within a small k radius around those points (see Fig. 7b, c, and Supplementary Tables 4, 5, and 6), the coefficient of correlation increases to 0.863. These relatively unstable points were chosen for measurement in order to map a complete ring around the drumhead, but are difficult to measure due to the extreme variance in the region $k_z > 1.0$. With larger circuits with much higher N, for example N = 30, the region $k_z > 1.0$ becomes easier to measure due to reduced variance at higher N (see Fig. 5a.).

Machine learning assisted selection of sampling points: To minimize the effect of uncertainties and reduce the number of $(k_y, k_z)$ points needed to reconstruct a prominent drumhead region of elevated impedance, we used a Nearest-Neighbor machine learning algorithm to select $(k_y, k_z)$ sampling points which are optimally impervious to capacitor and inductor uncertainties, see Fig. 8. This is important for reducing experimental costs, as well as reducing the impact of inevitable component uncertainties.

We associate each sampling point with $Z_{avg}$, which is the impedance of a particular $k_y, k_z$ point averaged over a large number of randomly generated

component uncertainties within the tolerance range, $+/-1\%$ for both inductors and capacitors, and a fixed parasitic resistance of $R_{pL} = 0.11\ \Omega$, $R_{pC} = 0.03\ \Omega$. $Z_{SD}$ is the corresponding standard deviation associated with many samples of a particular $k_y, k_z$ point simulated with component uncertainty and parasitic resistance, and $Z_{ideal}$ is the predicted impedance without component uncertainty or parasitic resistance. To ensure the integrity of the measured data, we will first desire that $|Z_{avg} - Z_{ideal}|/|Z_{ideal}|$ is small. This is not necessarily the case when the impedance depends highly nonlinearly with the capacitances and inductances. Furthermore, $Z_{SD}/Z_{ideal}$ should be minimized too, so as to mitigate the variance caused by the uncertainties.

Given an initial set of sampling points, our selection algorithm improves on them according to the aforementioned metrics, and outputs a more desirable set of points. As elaborated in Fig. 8, the Nearest-Neighbor unsupervised learning algorithm efficiently determines a smaller allowed search space, allowing the filtering of desirable measurement points to be performed with much less computational resources compared to a brute force approach. We have optimized the selection of sampling points only within the drumhead region, since high impedance points are more sensitive to parasitic resistance and component uncertainty.

Tuning of each unit cell through variable inductors: To realize the Laplacian at a specific $(k_y, k_z)$ point, the admittance of each coupling unit, i.e., logical component $x$ must be tuned to correspond to $G_x$ in (18). This may be done by attaching each coupling unit to a voltage divider and AC power supply, and observing the voltage drop across the coupling unit using an oscilloscope, see Fig. 10. The coupling unit is placed in series with a calibrating resistance $R_t = 1.5$ k$\Omega$, a $V_{\text{supply}} = 5$ Vpp voltage is supplied across the entire circuit, and the voltage amplitude $V_x$ between the ends of the coupling unit is measured. The inductance of the variable inductor in each coupling unit shall be tuned until $V_x$ matches with

$$V_x = \frac{V_{\text{supply}}}{1 + R_t G_x(k_y, k_z)} \quad \text{for} \quad x \in [t, v, -t, t_{AB}, g_A, g_B]. \tag{19}$$

Each variable inductor is tuned using ferrite rods or shorted wire loops placed close to the fixed-value inductors, see Fig. 6. To increase the inductance, a ferrite rod is placed closer to the inductor to better align the internal magnetic fields in it. Conversely, to decrease the inductance, a wire loop is used to "shield" the inductor from any change in magnetic field, thereby decreasing its self-inductance. The wire loop decreases the inductance of the original fixed-value inductor due to an opposing induced current, as derived below. First, the e.m.f induced in the wire loop is equal to

$$\epsilon = -\frac{d}{dt}\Phi_{wl} = -j\omega L_m i(t), \tag{20}$$

where $L_m$ is the mutual inductance between the fixed-value inductor and the wire loop, and $i(t)$ is the current running through the fixed-value inductor. The current in the wire loop is then

$$i_{wl} = \frac{\epsilon}{Z_{wl}} = \frac{-j\omega L_m i(t)}{j\omega L_{wl} + R_{wl}}, \tag{21}$$

where $L_{wl}$ is the self-inductance of the wire loop, and $R_{wl}$ is the resistance of the wire loop. At sufficiently high AC frequencies, we may ignore the resistance in the wire loop, such that the current induced in the wireloop is simply

$$i_{wl}(t) \approx -\frac{L_m}{L_{wl}}i(t). \tag{22}$$

The total flux on the original fixed-value inductor is then

$$\Phi = Li(t) + L_m i_{wl}(t) \approx \left(L_i - \frac{L_m^2}{L_{wl}}\right)i(t), \tag{23}$$

implying a decreased inductance in the original fixed-value inductor:

$$L = \frac{\Phi}{i(t)} \approx L_i - \frac{L_m^2}{L_{wl}}. \tag{24}$$

Using a combination of the ferrite rod and wire cage, we were able to alter the inductance of a fixed-value inductor component by $-50\%$ to $+25\%$ of its original manufactured value.

With this range of variable inductances and the known stipulated values of the logical components given in Supplementary Table 1, we selected default fixed inductor values of 39 uH for the $t$, $v$ coupling units, and 10 uH for the remaining $-t, t_{AB}, g_A, g_B$ units. The default capacitor values were selected to reproduce the reference point $(k_y, k_z) = (1.02, 0.75)$ via ((18)) without any alteration of the fixed-value inductors. Because capacitors are only sold in a restricted set of standard values, we used a parallel combination of several standard capacitors to make up the capacitances needed in all of the coupling units. The combinations used in the experiment for the coupling units $t$, $v$, $-t$, $t_{AB}$, $g_A$, $g_B$ are shown in Supplementary Table 1. In addition to the variable inductors, removable 470 or 1800 pF capacitors are sometimes connected in parallel to certain coupling units to represent $(k_y, k_z)$ points beyond the tuning range of the variable inductors alone. See Supplementary Table 2 for a complete list of component part numbers used in the experiment.

Offsetting calibration uncertainty: In the experiment, all variable inductor values are calibrated by a voltage divider as illustrated by Fig. 6d. As suggested by (19), they are crucially dependent on the known value of the calibrating resistance $R_t$. In particular, suppose that $R_t$ has a manufacturing uncertainty $\Delta R_t$. Then since the calibration voltage depends only on the product $R_t G_x$, the admittance of component $G_x$ will also sustain a measurement error of $\Delta G_x / G_x = -\Delta R_t / R_t$. Since $G_x$ is related to the frequency via $G_x = i\omega C_x + (i\omega L_x)^{-1}$ in (18), the effect of a nonzero $\Delta R_t$ can be offset by shifting the measurement frequency window by

$$\begin{aligned}
\Delta \omega &\approx \frac{\Delta G_x}{dG_x/d\omega} \\
&= -\frac{G_x \Delta R_t}{R_t dG_x/d\omega} \\
&= -\frac{i\omega C_x + \frac{1}{i\omega L_x}}{iC_x - \frac{1}{i\omega^2 L_x}}\frac{\Delta R_t}{R_t} \\
&= -\omega \frac{\omega^2 - \omega_x^2}{\omega^2 + \omega_x^2}\frac{\Delta R_t}{R_t},
\end{aligned} \tag{25}$$

where $\omega_x^2 = (L_x C_x)^{-1}$ is the resonant frequency of coupling unit $x$. As such, calibration uncertainties can be offset by a small shift (in this case empirically determined to be all close to $-60$ kHz) in the measurement frequency up to leading order, allowing the drumhead region to still be faithfully mapped out.

## Data availability

The data that support the findings of this study are available from the corresponding author upon reasonable request.

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

## Acknowledgements
The work in Würzburg is funded by the Deutsche Forschungsgemeinschaft (DFG, German Research Foundation) through Project-ID 258499086 - SFB 1170 and through the Würzburg-Dresden Cluster of Excellence on Complexity and Topology in Quantum Matter – *ct.qmat* Project-ID 39085490 - EXC 2147. T. Helbig was supported by a Ph.D. scholarship of the Studienstiftung des deutschen Volkes, Germany. X.Z. is supported by the National Natural Science Foundation of China (Grant No. 11874431), the National Key R&D Program of China (Grant No. 2018YFA0306800), and the Guangdong Science and Technology Innovation Youth Talent Program (Grant No. 2016TQ03X688). A.S., Y.S.A., and L.K.A. are supported by A*STAR-IRG (A1783c0011) and Singapore Ministry of Education (MOE) Academic Research Fund (AcRF) Tier 2 grant (2018-T2-1-007). Open access funding provided by Projekt DEAL.

## Author contributions
C.H.L. conceptualized and initiated the project, designed the experiment and wrote most of the manuscript. T. Hofmann and T. Helbig performed the numerical simulations and provided circuit expertise. Y.L. and X.Z. provided support on the mathematical aspects. A.S. performed the experiment under the guidance of Y.S.A. L.K.A., M.G. and R.T. took on advisory roles and wrote parts of the manuscript. The manuscript reflects the contributions of all authors.

## Competing Interests
The authors declare no competing interests.
