## [Peer Review File · Nature Communications]

Reviewers' comments:

Reviewer #1 (Remarks to the Author):

The authors theoretically propose how to realize nodal-knot structure in electric circuits. Some of the authors are pioneers of topoelectric circuits. Now, it is well established that almost all of the non-interacting-particle tight-binding Hamiltonian in condensed matter physics correspond to adequately designed equipped electric circuits. In this context, it is not surprising that nodal knots are realized in electric circuits. I understand that "our key insight is to instead realize pairs of nodal knots related by mirror symmetry, such that reciprocity does not have to be broken." is the major advance found in this paper. However, it is not a significant advance which warrants the publication in Nature Communications. It is a rather technical advancement which may be suitable for more specialized journal. In this sense, I cannot recommend it for Nature Communications. On the other hand, provided this work is submitted together with experimental results, it will be reconsidered in Nature Communications. I believe that experiments are possible for them since some of the authors already reported some nice experiments on topoelectric circuits in a previous work.

Here are specific comments, which should be properly answered when the authors resubmit or transfer the manuscript.

1)

The authors mention the RLC circuits. However, in my understanding, resistors are not considered in this work. Note that resistors induce non-Hermitian effects. See M. Ezawa, Non-Hermitian higher-order topological states in nonreciprocal and reciprocal systems with their electric-circuit realization Phys. Rev. B 99, 201411(R) (2019)

The authors should discuss the non-Hermitian effect induced by the resistors since the resistance always exists in actual electric circuits.

Are the results robust in the presence of the non-Hermiticity?

2)

The authors should explicitly write down the circuit Laplacian. The corresponding Hamiltonian is not enough.

3)

I wonder how to measure the Fourier transformed two-point impedance. We note that we need two points a and b to obtain the impedance. On the other hand, eq.(10) contains only one variable, which is peculiar. For example, what is the Fourier transformed form of eq.(8)? In realistic experiments, I think the measurement of the Fourier transformed two-point impedance in 3D is almost impossible.

4)

With respect to the statement " the effective density of states [17] and even admittance band structure [15, 18] can be obtained with just impedance and voltage/current measurements, respectively", I wonder if the effective DOS were really discussed in Ref.17. What is the meaning of "effective"? Is the DOS a local DOS or the total DOS? How to measure the DOS by the impedance? The authors should discuss the DOS measurement of the nodal knots in more details in this paper.

5)

The explanations of the black dots in Figs.3 and 4 are not clear enough. The authors should add more explanations. What is the admittance threshold?

6)

I wonder why the authors use various system sizes in this work. Is it possible use the same system size for all calculations?

7)

The reason of the choice of the $f(z,w)$ for the Figure-8 knot circuit is not clear. How can we obtain this explicit form? I understand that all of the knot structures are automatically constructed by general Hamiltonian, which is proposed in the context of topological Hopf semimetals.

For example, see Ref.19.

The authors should comment on the general formula generating knot structure with arbitrary large Hopf number.

8)

The reason of the choice of the coefficient 1.15, 0.1, 1.25 and -2.1 in eq.(15) is not clear. How is it robust for the choice of these coefficients?

For example, is it enough to use 1, 0, 1 and -2?

It is important since we cannot tune the variable with arbitrary accuracy in realistic experiment since the values of L and C are discrete for practical samples.

9)

The f proposed in the page 10 and 11 will be very hard to implement in experiments. Is it possible to simplify them for experimentalists?

10)

The authors mainly cite only their own papers on topolelectric circuits. Now, there are some important papers on topolelectric circuits by other authors. It is unfair to neglect all of these, although I admit that this field was initiated by some of the authors. The authors should cite more references on topolelectric circuits.

11)

The authors should show the details of Xyce simulations so that the readers can reproduce the results.

Reviewer #2 (Remarks to the Author):

In the manuscript titled "Imaging nodal knots in momentum space through topolelectrical circuits", Lee et al put forward a nice approach to construct nodal knots in momentum space. Nodal-link semimetals and nodal-knot semimetals with links and knots in the momentum space have been an active research topic in recent years. There is, however, no physical realization of these novel topological semimetals in realistic systems. To proceed in this research direction, a physical realization is highly desirable. As far as I know, the present manuscript provides the first realistic proposal for nodal knot in momentum space. The topolelectrical circuits are ideal platform for their realization, allowing a complete physical characterization. The idea is carried out in an in-depth way, and the manuscript is well written, therefore, I strongly recommend its publication in Nature Communications.

I have several suggestions for further improvement of the manuscript:

1) It is useful to add a calculation/discussion about the topological transition points where a ring crosses with other ring or itself. There should be singularity in measurable quantity. It is useful to calculate it, or at least, add a discussion about it.

2) The topoelectrical circuit is an ideal platform for studying many topological phases. One particular advantage of topoelectrical circuits is that non-Hermiticity can be readily added. It would be useful to add a few remarks about this.

3) A few relevant references about nodal-link semimetals can be added: Yan et al, Phys. Rev. B 96, 041103; Chen et al, Phys. Rev. B 96, 041102.

Reviewer 1

“It is well established that almost all of the non-interacting-particle tight-binding Hamiltonian in condensed matter physics correspond to adequately designed equipped electric circuits.”

We agree with Reviewer 1 that indeed, circuit platforms are extremely versatile, especially for realizing lattices with complicated long-ranged couplings. That said, if one were to restrict circuits to only reciprocal RLC elements, which are the ones that are far easier to build and troubleshoot, it becomes a non-trivial problem to design various kinds of band structures i.e. 3D nodal knots whose set of 2D slices can represent Chern number transitions necessitating non-reciprocity i.e., time-reversal breaking.

“I understand that “our key insight is to instead realize pairs of nodal knots related by mirror symmetry, such that reciprocity does not have to be broken.” is the major advance found in this paper. However, it is not a significant advance which warrants the publication in Nature Communications. It is a rather technical advancement which may be suitable for more specialized journal.”

Although our proposal of circumventing the requirement of reciprocity breaking by realizing pairs of nodal knots may appear like a technical advancement, we would like to stress its fundamental significance to the referee in the following.

One of the holy grails in the nodal semimetal community is to find feasible realizations of various nodal knots and links. These realizations also possess the wider significance of accomplishing sophisticated topological objects far more esoteric than the usual Chern or Z_2 topological insulators, or Dirac/nodal loop semimetals. So far, nodal knots have never been experimentally realized; our work communicates that circuits are the most promising platform, a perspective also shared by Reviewer 2. For this purpose, the main bottleneck is to design a circuit network such that it can be conveniently realized with LC components, which

are far simpler to work with than non-reciprocal components whose large-scale fine-tuning of feedback is nontrivial. The majority of topological circuits experiments so far had been done with reciprocal rather than non-reciprocal components. As such, the appeal of our work to the community is also evidenced in its 13 citations garnered even as a preprint, as reported by the NASA ADS database.

In fact, the accessibility of our fully reciprocal design is now demonstrated through our experiment, which involves only LC components. By contrast, another experimental work on a mathematically simpler but non-reciprocal system by some of the authors of this work [Helbig et al., arXiv:1907.11562] which is under the second round of review by Nature Physics and requires much more precision tuning, and is reported as a full-fledged experimental paper.

“On the other hand, provided this work is submitted together with experimental results, it will be reconsidered in Nature Communications. I believe that experiments are possible for them since some of the authors already reported some nice experiments on topolectric circuits in a previous work” ... “Some of the authors are pioneers of topolectric circuits”.

We thank Reviewer 1 for assuring his belief in the impact of our work such as to agree to reconsider it for Nature Communications if we resubmit with experimental results. We also thank him/her for recognizing our experience in the field, and for having the trust in us that we are able to perform an experimental demonstration of our work.

Over the last few months, we have managed to measure arguably the most interesting aspect of nodal structures: their topological drumhead states which, if repeated for surface terminations in different directions, will allow the reconstruction of the 3D nodal structure. We are very glad to have been able to perform this experiment – which includes resource and manpower acquisition, experimental design, simulation, prototyping, PCB manufacturing and finally experimental measurement – in the span of less than half a year, including manufacturer delays over Christmas and New Year seasons, notwithstanding delays due to the Coronavirus. While it would have been unrealistic to perform a full-fledged experiment where we fabricate and measure a fully 3D multi-knot circuit realization in a timeframe reasonable for a journal resubmission, the Hopf link we have achieved in hybrid synthetic and physical space does confirm our theoretical predictions very well, and provides the proof-of-principle for having a purely reciprocal circuit realization.

Our experiment exceeds the sophistication of most existing experimental works on topological circuits: The accurate sampling across synthetic space had required a precision of tuning unprecedented in the field of topolectrical circuits, which we are proud to have demonstrated in this experiment. Furthermore, this tuning has been done in a physical lattice more complicated than existing experiments. As a consequence, our experiment does not just demonstrate the feasibility of using our fully-reciprocal approach for nodal structure engineering, but has also advanced the state-of-the-art for accurate tuning of circuit lattices, where maintaining uniformity across unit cells is a nontrivial challenge. As such, we have also pedagogically detailed our experimental procedure at length.

1. *“The authors mention the RLC circuits. However, in my understanding, resistors are not considered in this work. Note that resistors induce non-Hermitian effects. See ‘M. Ezawa, Non-Hermitian higher-order topological states in nonreciprocal and reciprocal systems with their electric-circuit realization, Phys. Rev. B 99, 201411(R) (2019)’. The authors should discuss the non-Hermitian effect induced by the resistors since the resistance always exists in actual electric circuits. Are the results robust in the presence of the non-Hermiticity?”*

We thank Reviewer 1 for raising this important point about resistors and the non-hermiticity it induces, and for providing a nice reference pertaining to it. The effects of finite resistivity are also present in our experiment and are, as we had demonstrated, not sufficient to damage the nodal drumhead states and their associated nodal structures.

We agree with the referee, that any realistic circuit setup involves the presence of resistances, be it through parasitic serial resistances in inductive or capacitive elements, solder joints or other undesired effects introduced by the actual circuit board. Those types of passive resistances always result in non-Hermitian contributions, as the complex impedance for resistors is phase-shifted by $\pi/2$ with respect to that for inductors or capacitors. In the present circuit design, such a non-Hermiticity can be captured by complex-valued coefficients in the defining knot function.

At the nodal line of the admittance band structure, the two bands touch and the band gap closes. A non-Hermitian term added to such a band touching point results in complex eigenvalues, where the band touching splits up into complex eigenvalue degeneracies, referred to as exceptional points. Applying this scenario to a graphene band structure, the Dirac cone splits up into a ring of exceptional points centered around its original location upon introducing imaginary on-site potentials, which can be realized by gain and loss.

For the three-dimensional nodal line, similar non-Hermitian terms result in a splitting of the line into an exceptional tube wrapping around the original nodal line. As a result, the region of the Brillouin zone, for which small admittance eigenvalues occur, is enlarged by the splitting. This effect is limited and the nodal structure is preserved, as long as the distance of two points along the nodal line is larger than the imaginary term in the knot function relative to its real part. This is the general behaviour, if small parasitic resistances are considered as the main cause of non-Hermiticity.

However, such behavior can potentially make the mapping of the nodal knots from simulated or actual measurements more difficult, as described in [Helbig et al., PRB 99, 161114 (2019)], in the main text’s discussion, and in the responses to questions 3 and 6. If the exceptional tube at some point in the Brillouin zone comes too close to the vicinity at another point, the enlarged regions of small admittance eigenvalues merge at a mutual point, and the original nodal line cannot be recovered anymore. This is a consequence of the requirement for the reconstruction of the exact admittance band structure of the circuit system and the filtering of admittance eigenvalues close to zero below a certain threshold, as described below.

In our experiment, we therefore resort to measuring peak values in the two-point impedance in reciprocal space to resolve the drumhead states that emerge for open boundary conditions in one direction as a direct consequence of the nodal knot in the fully periodic system. Those impedance peaks are more robust to resistive perturbations, because their shape is preserved and only dampened in amplitude.

We have also cited the reference mentioned by the referee, as well as a few others on similar directions.

2. *“The authors should explicitly write down the circuit Laplacian. The corresponding Hamiltonian is not enough.”*

Our circuits have always been described in the Laplacian formalism since Eq 1. Still, for the benefit of clarity, we have showed how the specified $f(z,w)$ translates to the explicit Laplacian for the example of the Hopf link directly after Eq. 17.

3. *“I wonder how to measure the Fourier transformed two-point impedance. We note that we need two points a and b to obtain the impedance. On the other hand, eq.(10) contains only one variable, which is peculiar. For example, what is the Fourier transformed form of eq.(8)? In realistic experiments, I think the measurement of the Fourier transformed two-point impedance in 3D is almost impossible.”*

To describe the behavior of a circuit system, we use a formalism based on the grounded circuit Laplacian. Its definition and construction according to a given electrical network is detailed in [Lee et al., Comm. Phys. 1, 39 (2018)]. Based on this approach, we implement a periodic circuit with a two-node unit cell. According to translational invariance, the circuit lattice connectivity can be specified by a single variable \mathbf{r} , which denotes the unit cell spacing between two points. The real space structure of the resulting circuit Laplacian can further be diagonalized by a Fourier transformation, with only the sublattice degree of freedom remaining. The Fourier transformation via quasi-momentum \mathbf{k} is given by Eq. (10) in the main text, where the indices (i,j) refer to the sublattice structure in each unit cell.

In its simplest form, eq. (8) of the main text refers to the total impedance between two circuit lattice points a and b in real space, and results as a consequence of all lattice connections in the system. It can be represented as a sum of contributions from every eigenvalue and eigenvector of the Laplacian, labeled by an index λ .

In a general network, which can be periodic or otherwise, a and b in eq. (8) could represent quasi-momenta \mathbf{k} and \mathbf{k}' , where (8) determines the total impedance between those two points in the momentum lattice. The momentum impedance is equivalent to an index-wise Fourier transformation of the real space impedance to reciprocal space and retains two indices, consistent with this basis transformation. In a periodic circuit network, the momentum impedance simplifies to contributions from the Fourier eigenmodes for \mathbf{k} and \mathbf{k}' . The real space two-point impedance can be written as a Fourier superposition of all momentum contributions, and vice versa. To reconstruct a specific momentum contribution, one needs to apply a measurement procedure, which provides all linearly independent real space impedances in the periodic network, and to perform a Fourier transformation of those.

In the same way, such a measurement procedure, where one measures the global response of the circuit to a local excitation can be used to reconstruct all momentum modes of a system. This allows for the definition of an admittance band structure, which serves as a starting point of all investigations in our paper and is experimentally accessible. In a periodic circuit, the measurement procedure can be significantly simplified [Helbig et al., PRB 99, 161114

(2019)]. Given that it has already demonstrated the measurement of Fourier transformed two-point impedance in 2, an analogous measurement in 3D should present no additional fundamental difficulty.

To elaborate, the complexity of the measurement highly depends on the total number of unit cells and the boundary conditions of the system, and is therefore only indirectly related to the dimensionality of the circuit. In the performed simulations, we mimic those measurements by accessing the real space voltages to a pre-defined current excitation. In our performed experiment, we resort to the discussed momentum impedances and search for peaks in their profile, which delineate the position and existence of the nodal line in the Brillouin zone due to their inverse proportionality to admittance eigenvalues. The latter measurement is more robust to resistive perturbations, as detailed in our answer to question 1.

4. We thank Reviewer 1 for requesting to clarify this important point. Indeed, the reference to an “effective density of states” has been confusing, and we have accordingly modified the text. The key point here, and in Ref. 17, is that density of states divergences lead to many degenerate eigenvalues, which will necessarily lead to a stronger divergence in the impedance via Eq 8 when these eigenvalues can be tuned to zero, i.e., by suitable grounded connections. We have also clarified this point by adding a few new lines one paragraph after Eq. 8.

5. “The explanations of the black dots in Figs. 3 and 4 are not clear enough. The authors should add more explanations. What is the admittance threshold?”

We agree with the referee about the confusion in the captions of Fig. 3 and 4 concerning the explanation of the black dots and the admittance threshold. We accordingly modified this caption for better clarity. For a discussion on the admittance threshold, we refer to our answer to question 6.

6. “I wonder why the authors use various system sizes in this work. Is it possible use the same system size for all calculations?”

The nodal lines and knots, which we describe in our paper, trace out a zero-admittance line in the three-dimensional Brillouin zone. In a specific circuit setup, be it a simulation or an experiment, the number of real space nodes is always finite, resulting in a lattice structure with discrete voltage nodes.

A Fourier transformation produces the reciprocal (momentum) lattice, which scales analogously with the number of unit cells in real space. The momentum grid results in a discrete sampling of the admittance band structure in the Brillouin zone. Depending on the allowed values of momenta and the precise trajectory of the nodal line, the band structure is sampled close to, but not directly at the nodal line.

As the momentum points will never exactly hit a zero admittance value, we determine an admittance threshold, which defines an upper bound of admittance. It is chosen to be much smaller than the admittance bandwidth of the system. The corresponding momenta to all admittance eigenvalues below this threshold are plotted in the three-dimensional Brillouin zone and expected to resemble a discrete sampling of the nodal knot.

Intuitively, the sampling improves for larger system sizes. An alternative approach to increasing the number of unit cells in a system is to create several copies of the same setup, but with varying system sizes. Choosing the number of unit cells as co-primes of one another increases the sampling density of the combined momentum grid without the need for creating a very large system. It features a more realistic approach to resolving the nodal knot in an experimental setting. To account for the question by the referee, we decided to address it directly in the paper on the section on simulation details in Methods.

7. The rationale behind the form of $f(z,w)$ for the Figure-8 knot was derived and explained in Ref. 29, and we have accordingly cited it alongside a brief explanation in the subsection on it.

While the general approach towards constructing $f(z,w)$ for any desired knot is already explained in Eq. 3 and the paragraphs after it, with some examples given, we have left the technical details of $f(z,w)$ for non-torus knots like the Figure 8 knot to Ref 29, in order not to detract us from the main focus of this work. The main inspiration behind the $f(z,w)$ of the Figure 8 knot is it is a lemniscate knot, i.e. a knot where the corresponding braid consists of a permutation of the strands over 1 cycle. As such, its full braid has to be composed a few times consecutively. This breaks the holomorphic dependence on w , and thus both w and \bar{w} appears.

8. “The reason of the choice of the coefficient 1.15, 0.1, 1.25 and -2.1 in eq.(15) is not clear. How is it robust for the choice of these coefficients? For example, is it enough to use 1, 0, 1 and -2? It is important since we cannot tune the variable with arbitrary accuracy in realistic experiment since the values of L and C are discrete for practical samples.”

We thank the referee for this intriguing question and decided to expand on this point in the main text of our paper and explain our choice of the coefficients. There are several approaches on how to overcome the practical difficulties mentioned by the referee, which concern the experimental realization of the nodal knot circuits.

First, as suggested by the referee, a small variation in the prefactors of the cosine and sine functions in the definition of the knot function preserves the principal structure of the knot, while merely deforming its shape in the Brillouin zone. The robustness of a knot upon such a coefficient modification highly depends on its specific implementation, but in general turns out to be fulfilled, if the relative change in the coefficients is smaller than the minimal distance of two points along the nodal line as compared to the total admittance bandwidth. In this particular trefoil knot in question, the tolerance for most of the coefficient turns out to be about 20-30%, as we have now indicated in the text.

Second, a global prefactor to the Laplacian does not change the shape of its nodal line. For experimental implementations, one can consequently always scale the Laplacian such that all available components fit the circuit board design. One can furthermore tune the excitation frequency to an optimal value, and thus align the ratio between complex inductances and capacitances of pre-chosen circuit elements with the demands by the knot function.

Third, to realize non-integer values of inductances and capacitances beyond their commercial availability, one can combine multiple capacitors or inductors in a serial or parallel connection. With an organized basis set of distinct component values, one can accomplish a

multitude of different connectivities and reduce the requirement of having many distinct component values at use.

Regarding the simulations shown in our paper, we adjust the shape of the nodal knots to find a set of parameters optimizing the complexity of simulations against the resolution of the knot in the three-dimensional momentum grid given by the implemented system sizes.

As it becomes clear by our work, the actual experimental implementation of nodal knot circuits still requires a lot of engineering and fine-tuning work. We concentrated on the development of the knot theory in circuit realizations as well as a proof-of-principle realization and demonstration of the theoretical work in both simulations and experiment.

9. The $f(z,w)$ presented in our work are taken directly from a braid parametrization of the knots, in fact some exactly from Ref 29 which we cited. Since both braids and knots can be considerably deformed without changing their topology, these $f(z,w)$ are by no means unique.

Indeed, it is in general possible to simplify their tight-binding expression for experimental implementations. But the simplification contains a few caveats:

(1) Some long-range couplings must remain, in order to realize the requisite number of twists in momentum space for a particular knot. However, these long-range couplings are not a problem for circuits, since wires can easily connect nodes that are not neighbors.

(2) the exact criterion for what is the “simplest” depends on the experimental implementation, or even the parts available for purchase. For instance, it may be preferable to implement more connections per unit cell, if that can be printed on a chip, than an alternative with fewer connections but which involves fine-tuned capacitor/inductor values.

(3) For any implementation, it is crucial to have reasonable tolerance to inevitable component uncertainties. The exact susceptibility to the uncertainties depends in a complicated way on the circuit, and robust configurations can only be found numerically. Such optimization problems can also be tackled with machine learning approaches, as is done as a proof of concept in our experiment. Often, the simplest circuit configurations may involve small gaps which have poor tolerance.

Based on these considerations, further simplification of the tight binding expressions for the circuits will only be meaningful when optimized according to the exact experimental constraints/resources in mind. That said, an illustration of a possible simplification procedure is given by Eqs 25-27 for our Hopf link, whose experimental implementation was derived via simplifying the original mathematical model obtained from $f(z,w) = z^2 - w^2$.

10. We have accordingly cited several other relevant works on circuits, particularly those from other groups.

11. *“The authors should show the details of Xyce simulations so that the readers can reproduce the results.”*

We agree with the referee on this point and thank him for the suggestion. In the methods section, for this purpose, we explicitly state the complete Laplacians for all performed simulations, from which the circuit connectivity can be deduced in full detail. We furthermore added a paragraph on the precise simulation setup, which reveals the details on the used circuit components, implementation of the current excitation, external excitation frequency, and on all other simulation parameters.

We further provide full transparency in the data availability for the simulations as well as the experiment. The netlists, which specify the simulation parameters and which can be directly run with Xyce to reproduce our results are available from the corresponding author upon reasonable request. Those netlists also specify the electrical circuit components between all voltage nodes in the system, as can be read off from the Laplacian matrices in the methods.

“Lee et al put forward a nice approach to construct nodal knots in momentum space. Nodal-link semimetals and nodal-knot semimetals with links and knots in the momentum space have been an active research topic in recent years. There is, however, no physical realization of these novel topological semimetals in realistic systems. To proceed in this research direction, a physical realization is highly desirable. As far as I know, the present manuscript provides the first realistic proposal for nodal knot in momentum space. The topoelectrical circuits are ideal platform for their realization, allowing a complete physical characterization. The idea is carried out in an in-depth way, and the manuscript is well-written, therefore, I strongly recommend its publication in Nature Communications.”

We are thankful to Reviewer 2 for his/her very positive recommendation of our manuscript for publication, and the nice summary of the significance of our direction as well as how our work places itself therein. With the addition of an experimental measurement of the nodal drumhead states, we hope to further strengthen Reviewer 2’s conviction that our work is the first proposal that is indeed realistic for realizing such nodal structures in a versatile way that allows for “a complete physical characterization”.

1. *“It is useful to add a calculation/discussion about the topological transition points where a ring crosses with other ring or itself. There should be singularity in measurable quantity. It is useful to calculate it, or at least, add a discussion about it.”*

Unlike the band topology of insulators, whose winding properties result in quantities like the Chern number that is directly proportional to the Hall conductivity, the topology of nodal knots do not correspond directly to any measurable quantity.

That said, we can still indirectly map out the nodal topology via impedance measurements, for the case of a nodal circuit. The most robust way to map it out is via the drumhead states, which involve mapping out the surface states of a nodal circuit, as we have now done experimentally. Since the drumhead states are the “shadows” of the nodal structure, if a nodal ring were to cross another nodal ring, we will see two drumhead regions merging into one, or vice versa. In this sense, the topology (connectivity of manifold) of the drumhead state can change, as long as we are not looking at a surface projection where the drumhead states are degenerate. We added a short discussion on this in the subsection “Surface states of knots”.

2. The topoelectrical circuit is an ideal platform for studying many topological phases. One particular advantage of topoelectrical circuits is that non-Hermiticity can be readily added. It would be useful to add a few remarks about this.

Indeed, the referee is right, and any realistic circuit setup involves the presence of resistances, be it through resistors, parasitic serial resistances in inductive or capacitive elements, solder joints or other undesired effects introduced by the actual circuit board, which all lead to non-hermiticity. Those types of passive resistances always result in non-Hermitian contributions, as the complex impedance for resistors is phase-shifted by $\pi/2$ with respect to that for inductors or capacitors.

We have added some remarks about non-Hermiticity in our discussion section. The idea behind it is that, at a nodal line of the admittance band structure, the two bands touch and the band gap closes. A non-Hermitian term added to such a band touching point results in complex eigenvalues, where the band touching splits up into complex eigenvalue degeneracies, referred to as exceptional points. For the three-dimensional nodal line, non-Hermitian terms result in a splitting of the line into an exceptional tube wrapping around the original nodal line. As a result, the region of the Brillouin zone, for which small admittance eigenvalues occur, is enlarged by the splitting.

If one were to focus on the study of such non-Hermitian exceptional structures, the resistances can be deliberately kept large. But otherwise, keeping the resistances small (of the same order as what we had in our new experiment), such effects are limited and the nodal structure is preserved, as long as the distance of two points along the nodal line is larger than the imaginary term in the knot function relative to its real part.

3. We thank Reviewer 2 for reminding us about these pioneering works. We have accordingly cited them.

REVIEWERS' COMMENTS:

Reviewer #1 (Remarks to the Author):

In the revised manuscript, the authors have add experimental results according to my previous suggestion, which is satisfactory.

I recommend to publith the manuscript after minor corrections.

1)

Why are there white unfilled regions in Fig.12a? Does the impedance diverge at these regions?

2)

The authors should define "AAH" model in page 7. Is it a Aubry-Andre-Harper model?

3)

In page 11, "threedimensional" should be "three-dimensional"

4)

I have a comment with respect to the statement

"Requiring no manual control input i.e. visual reference to an oscilloscope, this approach can achieve drastic speedup for the tuning, and perhaps even allow real-time tuning for the simulation of Floquet Hamiltonians."

Could the authors add more discussions on Floquet Hamiltonians?

Is it related to the fact that the authors are applying alternating current to the circuit?

Reviewer #2 (Remarks to the Author):

With the added experiment and revisions, I think that all issues have been satisfactorily addressed. I would like to strongly recommend its acceptance in Nature Communications.

We are glad that both referees recommend publication.

Referee #1:

- 1) They are regions with impedance above 600 Ohms (red) in the figure, but do not diverge as the circuit components have nonvanishing impedance. We have clarified this in the caption.
- 2) Yes, and the abbreviation has been removed.
- 3) Corrected.

Referee #2:

We are glad that he/she has no further comments, and strongly recommends publication.